# 4K4DGen:
# Panoramic 4D Generation at 4K Resolution

**Renjie Li**[*,1,4], **Panwang Pan**[*†‡1], **Bangbang Yang**[*1], **Dejia Xu**[*2], **Shijie Zhou**[3], **Xuanyang Zhang**[1],
**Zeming Li**[1], **Achuta Kadambi**[3], **Zhangyang Wang**[2], **Zhengzhong Tu**[4], **Zhiwen Fan**[2]

[1]Bytedance, [2] University of Texas at Austin, [3] University of California, Los Angeles,
[4] Texas A&M University

**https://4k4dgen.github.io/**

paulpanwang@gmail.com

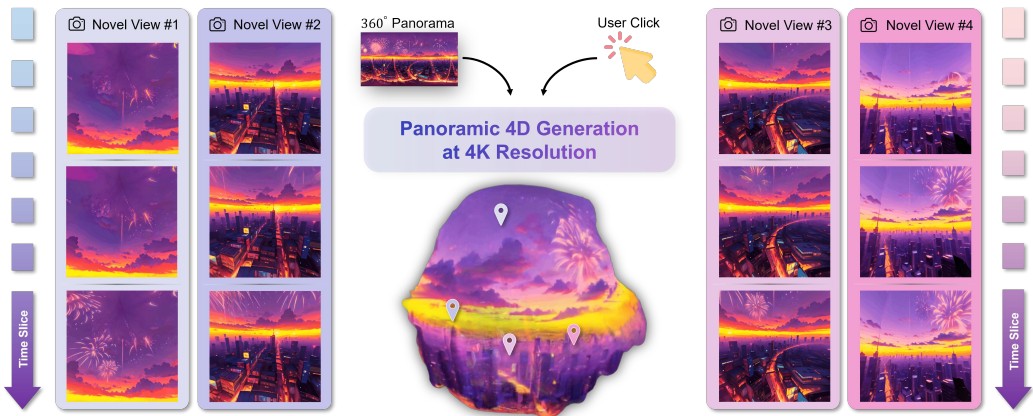

Figure 1: **4K4DGen** takes a static panoramic image with a resolution of $4096 \times 2048$ and allows animation through user interaction or an input mask, transforming the static panorama into dynamic Gaussian Splatting. 4K4DGen supports the rendering of novel views at various timestamps, enriching immersive virtual exploration.

## Abstract

The blooming of virtual reality and augmented reality (VR/AR) technologies has driven an increasing demand for the creation of high-quality, immersive, and dynamic environments. However, existing generative techniques either focus solely on dynamic objects or perform outpainting from a single perspective image, failing to meet the requirements of VR/AR applications that need free-viewpoint, 360° virtual views where users can move in all directions. In this work, we tackle the challenging task of elevating a single panorama to an immersive 4D experience. For the first time, we demonstrate the capability to generate omnidirectional dynamic scenes with 360° views at 4K ($4096 \times 2048$) resolution, thereby providing an immersive user experience. Our method introduces a pipeline that facilitates natural scene animations and optimizes a set of 3D Gaussians using efficient splatting techniques for real-time exploration. To overcome the lack of scene-scale annotated 4D data and models, especially in panoramic formats, we propose a novel **Panoramic Denoiser** that adapts generic 2D diffusion priors to animate consistently in 360° images, transforming them into panoramic videos with dynamic scenes at targeted regions. Subsequently, we propose **Dynamic Panoramic Lifting** to elevate the panoramic video into a 4D immersive environment while preserving spatial and temporal consistency. By transferring prior knowledge from 2D models in the perspective domain to the panoramic domain and the 4D lifting with spatial appearance and geometry regularization, we achieve high-quality Panorama-to-4D generation at a resolution of 4K for the first time.

---

∗: Equal contribution †: Project lead; ‡: Corresponding author.

# 1 INTRODUCTION

With the increasing growth of generative techniques (Rombach et al., 2022; Blattmann et al., 2023a), the capability to create high-quality assets has the potential to revolutionize content creation across VR/AR and other spatial computing platforms. Unlike 2D displays such as smartphones or tablets, ideal VR/AR content must deliver an immersive and seamless experience, enabling 6-DoF virtual tours and supporting high-resolution 4D environments with omnidirectional viewing capabilities. Despite significant advancements in the generation of images, videos, and 3D models, the development of panoramic 4D content has lagged, primarily due to the scarcity of well-annotated, high-quality 4D training data. Even in the most relevant field of 4D generation, existing works mainly focus on generating or compositing object-level contents (Bahmani et al., 2024; Lin et al., 2024), which are often in low-resolution (e.g., below 1080p) and cannot fulfill the demand of qualified immersive experiences. Based on these observations, we propose that an ideal generative tool for creating immersive environments should possess the following properties: **(i)** the generated content should exhibit high perceptual quality, reaching high-resolution (4K) output with dynamic elements (4D); **(ii)** the 4D representation must be capable of rendering coherent, continuous, and seamless 360° panoramic views in real time, supporting efficient 6-DoF virtual tours. However, creating diverse, high-quality 4D panoramic assets presents two significant challenges: **(i)** the scarcity of large-scale, annotated 4D data, particularly in panoramic formats, limits the training of specialized models. **(ii)** achieving both fine-grained local details and global coherence in 4D and 4K panoramic views is difficult for existing 2D diffusion models. These models, typically trained on perspective images with narrow fields of view (FoV), cannot be easily adapted to the expansive scopes of large panoramic images (see Exp. 4.3). On another front, video diffusion models (An et al., 2023) trained with web-scale multi-modal data have demonstrated versatile utility as region-based dynamic priors, and Gaussian Splatting (Kerbl et al., 2023) has shown efficient capabilities in modeling 4D environment. Thus, we address the large-scale, omnidirectional dynamic scene generation (4D panoramic generation) problem by utilizing the generative power of diffusion models to animate static panoramic images, transforming them into realistic, dynamic scenes that can support immersive, 360° viewing experiences. To achieve this, we propose to elevate the dynamic panoramic video to 4D environment assets using a set of dynamic Gaussians, which can be seamlessly integrated into VR/AR platforms for real-time rendering and interaction.

In this paper, we introduce **4K4DGen**, a novel framework designed to enable the creation of panoramic 4D environments at resolutions up to 4K. 4K4DGen addresses the key challenges of maintaining consistent object dynamics across the entire 360° field-of-view (FoV) in panoramic videos, while preserving both spatial and temporal coherence as the video transitions into a fully interactive 4D environment. Specifically, we propose the **Panoramic Denoiser**, which animates 360° FoV panoramic images by denoising spherical latent codes corresponding to user-interacted regions. The Panoramic Denoiser leverages a well-trained diffusion model originally designed for narrow-FoV perspective images, enabling the generation of 360° dynamic panoramas while ensuring global coherence and continuity throughout the entire panorama. To transform the omnidirectional panoramic video into a 4D environment, we introduce **Dynamic Panoramic Lifting**, which corrects scale discrepancies using a depth estimator enriched with perspective prior knowledge to generate panoramic depth maps. Additionally, it employs time-dependent 3D Gaussians optimized with spatial-temporal geometry alignment to ensure cross-frame consistency in dynamic scene representation and rendering. By adapting generic 2D statistical patterns from the perspective domain to the panoramic format and effectively regularizing Gaussian optimization with geometric principles, we achieve high-quality 4K panorama-to-4D content generation with photorealistic novel-view synthesis capabilities. Our contributions can be summarized as follows.

- We introduce **4K4DGen**, the first framework capable of generating high-resolution (up to 4096×2048) 4D omnidirectional assets without the need for annotated 4D data.

- We propose the **Panoramic Denoiser**, which transfers generative priors from pre-trained 2D perspective diffusion models to the panoramic space, enabling consistent animation of panoramas with dynamic scene elements.

- We introduce **Dynamic Panoramic Lifting**, a method that transforms dynamic panoramic videos into dynamic Gaussians, incorporating spatial-temporal regularization to ensure cross-frame consistency and coherence.

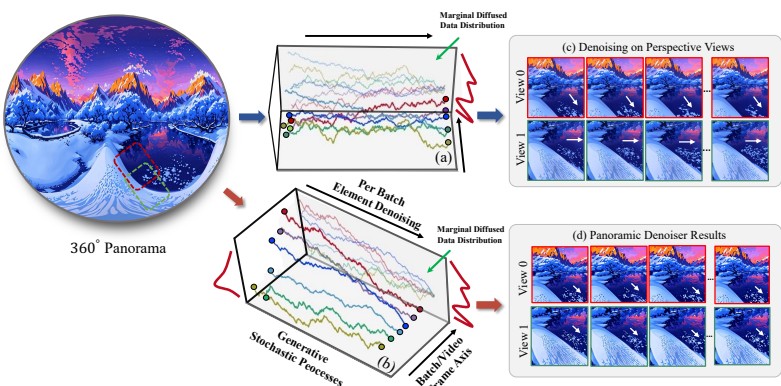

Figure 2: **Panoramic Denoiser** adapts diffusion priors from the perspective domain to the panoramic domain by simultaneously denoising perspective views and integrating them into spherical latents at each denoising step. This approach ensures consistent animation across multiple views.

## 2 RELATED WORK

**Diffusion-based Image and Video Generation.** Recent advancements have significantly expanded the capabilities of generating 2D images using diffusion models, as evidenced in several studies (Dhariwal & Nichol, 2021; Nichol et al., 2021; Podell et al., 2024; Ramesh et al., 2022; Saharia et al., 2022). Notably, Stable Diffusion (Rombach et al., 2022) optimizes diffusion models (DMs) within the latent spaces of autoencoders, striking an effective balance between computational efficiency and high image quality. Beyond text conditioning, there is increasing emphasis on integrating additional control signals for more precise image generation (Mou et al., 2024; Zhang et al., 2023). For example, ControlNet (Zhang et al., 2023) enhances the Stable Diffusion encoder to seamlessly incorporate these signals. Furthermore, the generation of images with consistent perspective views is gaining attention, such as the training-based techniques like (Tang et al., 2023; Höllein et al., 2024), or the sampling-based techniques like (Song et al., 2023; Bar-Tal et al., 2023; Lee et al., 2023; Quattrini et al., 2025). Diffusion models are also extensively applied in video generation, as demonstrated by various recent works (Ge et al., 2023; Ho et al., 2022; Wang et al., 2023a; Wu et al., 2023b; 2024b; Zhou et al., 2022). For instance, Imagen Video (Ho et al., 2022) utilizes a series of video diffusion models to generate videos from textual descriptions. Similarly, Make-A-Video (Singer et al., 2023) advances a diffusion-based text-to-image model to create videos without requiring paired text-video data. MagicVideo (Zhou et al., 2022) employs frame-wise adaptors and a causal temporal attention module for text-to-video synthesis. Video Latent Diffusion Model (VLDM) (Blattmann et al., 2023b) incorporates temporal layers into a 2D diffusion model to generate temporally coherent videos.

**3D/4D Large-scale Generation.** In recent 3D computer vision, a large-scale scene is usually represented as implicit or explicit fields for its appearance (Mildenhall et al., 2020; Kerbl et al., 2023), geometry (Peng et al., 2020; Wang et al., 2023b; Huang et al., 2023), and semantics (Kerr et al., 2023; Zhou et al., 2024a; Qin et al., 2024). We mainly discuss the 3D Gaussian Splatting (3DGS) (Kerbl et al., 2023) based generation here. Several works including DreamGaussian (Tang et al., 2024), GaussianDreamer (Yi et al., 2024), GSGEN (Chen et al., 2023), CG3D (Vilesov et al., 2023), and DiffSplat (Lin et al., 2025) employ 3DGS to generate diverse 3D objects and lay the foundations for compositionality, while LucidDreamer (Chung et al., 2023), Text2Immersion (Ouyang et al., 2023), GALA3D (Zhou et al., 2024c), RealmDreamer (Shriram et al., 2024), and DreamScene360 (Zhou et al., 2024b) aim to generate static large-scale 3D scenes from text. Considering the current advancements in 3D generation, investigations into 4D generation using 3DGS representation have also been conducted. DreamGaussian4D (Ren et al., 2024) accomplishes 4D generation based on a reference image. AYG (Ling et al., 2023) equips 3DGS with dynamic capabilities through a deformation network for text-to-4D generation. Besides, Efficient4D (Pan et al., 2024) and 4DGen (Yin et al., 2023) explore video-to-4D generation, and utilize SyncDreamer (Liu et al., 2023) to produce multi-view images from input frames as pseudo ground truth for training a dynamic 3DGS. 4K4D (Xu et al., 2024) is a high-resolution reconstruction technique that extends 3DGS to model complex human motion with detailed backgrounds while achieving real-time rendering speed.

**Panoramic Representation.** A panorama is an image that captures a wide, unbroken view of an area, typically encompassing a field of vision much wider than what a standard photo would cover, providing a more immersive representation of the subject. Recently, novel view synthesis using panoramic representation has been widely explored. For instance, PERF (Wang et al., 2024a) trains a panoramic neural radiance field from a single panorama to synthesize 360° novel views. 360Roam (Huang et al., 2022) proposed learning an omnidirectional neural radiance field and progressively estimating a 3D probabilistic occupancy map to speed up volume rendering. OmniNeRF (Gu et al., 2022) introduced an end-to-end framework for training NeRF using only 360° RGB images and their approximate poses. PanoHDR-NeRF (Gera et al., 2022) learns the full HDR radiance field from a low dynamic range (LDR) omnidirectional video by freely moving a standard camera around. In the realm of 3DGS, 360-GS (Bai et al., 2024) takes 4 panorama images and 2D room layouts as scene priors to reconstruct the panoramic Gaussian radiance field. DreamScene360 (Zhou et al., 2024b) achieves text-to-3D Panoramic Gaussian Splatting by utilizing monocular depth priors to regularize the Gaussian optimization.

# 3  METHODOLOGY

Taking a single panoramic image as input, the goal of 4K4DGen is to generate a panoramic 4D environment capable of rendering novel views from arbitrary angles and at various timestamps. Our approach initially constructs a panoramic video and then elevates it into a series of 3D Gaussians, enabling efficient splatting for flexible rendering. Naïvely animating projected perspective images, however, often results in unnatural motion and inconsistent animations. To overcome this, our method propose the denoising of projected spherical latents, ensuring consistent animation of the panoramic video from the original image, as detailed in Sec. 3.3.

Moreover, directly converting multiple perspective images from different timestamps into 4D frequently leads to degraded geometry and visible artifacts (see Sec. 4.3). We address this by applying spatial-temporal geometry fusion to lift the panoramic video, as described in Sec. 3.4. The complete pipeline of 4K4DGen is illustrated in Fig. 3.

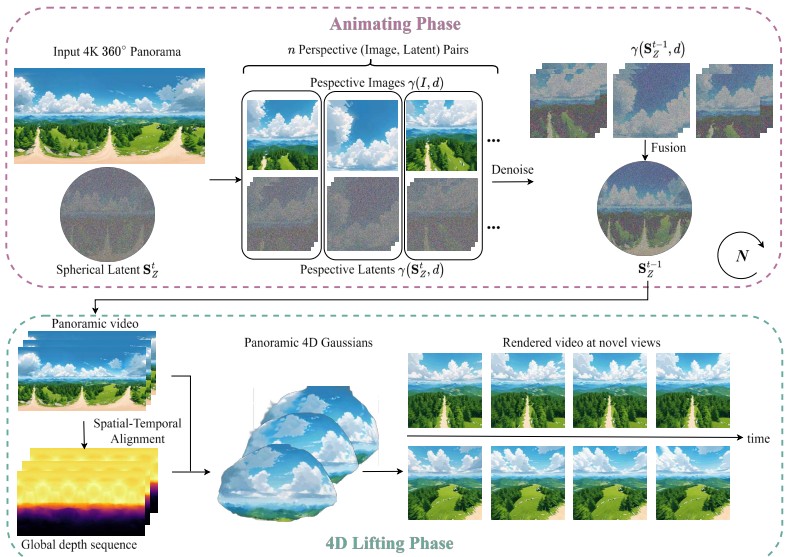

Figure 3: **Overall Pipeline.** Beginning with a static panorama as input, the **Animating Phase** generates a panoramic video by first mapping the panorama into a spherical latent space, followed by denoising within the perspective space, fusing back to the spherical latent space at each step, and finally transforming it into the panoramic space. In the **4D Lifting Phase**, a series of dynamic Gaussians is employed to lift the panoramic video into a 4D representation, ensuring both spatial and temporal consistency.

### 3.1 PRELIMINARIES

**Latent Diffusion Models (LDMs).** LDMs (Rombach et al., 2022) consist of a forward procedure $q$ and a backward procedure $p$. The forward procedure gradually introduces noise into the initial latent code $x_0 \in \mathbb{R}^{h \times w \times c}$, where $x_0 = \mathcal{E}(I)$ is the latent code of image $I$ within the latent space of a VAE, denoted by $\mathcal{E}$. Given the latent code at step $t - 1$, the $q$ procedure is described as $q(x_t | x_{t-1}) = \mathcal{N}(x_t; \sqrt{1 - \beta_t} x_{t-1}, \beta_t \boldsymbol{I})$. Conversely, the backward procedure $p$, aimed at progressively removing noise, is defined as $p_\theta(x_{t-1} | x_t) = \mathcal{N}(\mu_\theta(x_t, t), \Sigma_\theta(x_t, t))$. In practical applications, images are generated under the condition $y$, by progressively sampling from $x_T$ down to $x_0$. Recently, image-to-video (I2V) generation has been realized (Guo et al., 2024; Dai et al., 2023) by extending the latent code with an additional frame dimension and performing decoding at each frame. The denoising procedure is succinctly represented as $x_{t-1} = \Phi(x_t, I)$, where $x_t, x_{t-1} \in \mathbb{R}^{l \times h \times w \times c}$ represent the sampled latent codes and $I$ the conditioning image. Recently, image-to-video (I2V) generation has been achieved (Guo et al., 2024; Dai et al., 2023) by extending the latent code with an additional frame dimension and performing decoding at each frame. The denoising procedure is succinctly expressed as $x_{t-1} = \Phi(x_t, I)$, where $x_t, x_{t-1} \in \mathbb{R}^{l \times h \times w \times c}$ represent the sampled latent codes, and $I$ represents the conditioning image.

**Omnidirectional Panoramic Representation.** Panoramic images or videos, denoted as $I$, are typically represented using equirectangular projections, forming an $H \times W \times C$ matrix, where $H$ and $W$ denote the image resolution and $C$ represents the number of channels. While this format preserves the matrix structure, making it consistent with planar images captured by conventional cameras, it introduces distortions, especially noticeable near the polar regions of the projection. To mitigate these distortions, we adopt a spherical representation for panoramas, where pixel values are defined on a sphere $\mathbb{S}^2 = \{ \boldsymbol{d} = (x, y, z) | x, y, z \in \mathbb{R} \wedge |\boldsymbol{d}| = 1 \}$. For a more precise definition of the projection, we represent matrix-like images using a mapping $\mathcal{E}_I : [-1, 1]^2 \to \mathbb{R}^C$, which normalizes the image coordinates into the range $[0, 1]$. Thus, for any given pixel $(x, y) \in [-1, 1]^2$, the corresponding pixel value is determined by $\mathcal{E}_I(x, y)$. We define the spherical representation of panoramas using the field $\mathcal{S}_I : \mathbb{S}^2 \to \mathbb{R}^C$, where $\mathcal{S}_I(\boldsymbol{d})$ gives the pixel value at a given direction $\boldsymbol{d} = (x, y, z)$. The relationship between the spherical and equirectangular representations is established through the following projection formula:

$$\mathcal{S}_I(x, y, z) = \mathcal{E}_I \left( \frac{1}{\pi} \arccos \frac{y}{\sqrt{1 - z^2}}, \frac{2}{\pi} \arcsin z \right). \tag{1}$$

For perspective images, we define a virtual camera centered at the origin. The rays for each pixel are determined through ray casting, as described in (Mildenhall et al., 2020), where each ray $\boldsymbol{d}$ is represented by $\boldsymbol{r}(x, y, f, \boldsymbol{u}, \boldsymbol{s}, R)$. This representation takes into account the focal length $f$, the z-axis direction $\boldsymbol{u}$, the image plane size $\boldsymbol{s}$, and the camera's rotation along the z-axis $R$. Consequently, for a given panorama $I$, the perspective image $P$ can be projected using these camera parameters $(f, \boldsymbol{u}, \boldsymbol{s}, R)$ as:

$$\mathcal{E}_P(x, y) = \mathcal{S}_I \circ \boldsymbol{r}(x, y, f, \boldsymbol{u}, \boldsymbol{s}, R). \tag{2}$$

In this paper, we fix the focal length $f$, the image plane size $\boldsymbol{s}$, and the rotation $R$. We denote the process of projecting the panorama $I$ into a perspective image $i$, based on the camera's z-axis direction $\boldsymbol{u}$, as $i = \gamma(I, \boldsymbol{u})$.

### 3.2 INCONSISTENT PERSPECTIVE ANIMATION

Large-scale pre-trained 2D models have shown remarkable generative capabilities in creating images and videos, benefiting from vast multi-modal training data gathered from the Internet. However, acquiring high-quality 4D training data is considerably more challenging, and no current 4D dataset reaches the scale of those available for images and videos. Therefore, our approach aims to utilize the capabilities of video generative models to produce consistent panoramic 360° videos, which are then elevated to 4D. Nonetheless, the availability of panoramic videos is significantly more limited compared to planar perspective videos. Consequently, mainstream image-to-video (I2V) animation techniques may not perform optimally for panoramic formats, and the resolution of the videos remains constrained, as illustrated in Fig. 5 (b) and Tab. 2. Alternatively, the animator can be applied to perspective images. but this introduces inconsistencies across different projected views, as depicted in Fig. 5 (c)

## 3.3 Consistent Panoramic Animation

Limited by the scarcity of 4D training data in panoramic format, and given that large diffusion models are primarily trained on planar perspective videos, directly applying 2D perspective denoisers presents challenges in generating seamless panoramic videos with proper equirectangular projection, due to inconsistent motion across different views and the domain gap between spherical and perspective spaces. This constraint has driven us to develop a panoramic video generator in spherical space that leverages priors from general image-to-video (I2V) animation techniques, as shown in Fig. 2. Consequently, starting from a static input panorama, we animate it into a panoramic video, as demonstrated in the "Animating Phase" section of Fig. 3.

**Spherical Latent Space.** To generate panoramic video from a static panorama, we build up the denoise-in-latent-space schema (An et al., 2023; Blattmann et al., 2023a; Dai et al., 2023) in a spherical context. For general video generation, a noisy latent sample is progressively denoised using DDPM (Ho et al., 2020), conditioned on a static input image, and subsequently decoded into a video sequence by a pre-trained VAE decoder. However, in 4K4DGen, unlike the method for generating perspective planar videos, both the latent code and the static panorama input are represented on spheres. We start with the initial panoramic latent code $S^T : \mathbb{S}^2 \to \mathbb{R}^{L \times c}$, where $L$ denotes the number of video frames and $c$ the channels per frame. A novel Panoramic Denoiser is then applied to generate the clean panoramic latent code $S^0$, conditioned on the static input panorama $I \in \mathbb{R}^{H \times W}$. Subsequently, the equirectangular projection, as introduced in Sec. 3.1, projects the clean panoramic latent code into the matrix-like latent code $Z^0 \in \mathbb{R}^{h \times w \times L \times c}$, with $h$ and $w$ representing the resolution of the latent code. Each $k^{\text{th}}$ video frame $I^k$ in pixel space is decoded by the pre-trained VAE decoder as $I^k = \mathcal{D}(Z^0[:, :, k, :])$.

**Build the Panoramic Denoiser.** We leverage a pre-trained perspective video generative model (Dai et al., 2023) to build our Panoramic Denoiser. This video generator takes a perspective image $i \in \mathbb{R}^{p_H \times p_W \times c}$ and an initial latent code $z^T \in \mathbb{R}^{p_h \times p_w \times (L \times c)}$ as inputs, progressively denoising the latent code $z^T$ to a clean state $z^0$ through a denoising function $z^{t-1} = \Phi(z^t, i)$. Here, $p_h$ and $p_w$ represents the resolution of the latent code, $p_H$ and $p_W$ the resolution of the conditioning image, $c$ the number of channels, and $L$ the video length. Our goal is to transform the initial noisy panoramic latent code $S^T$ into the clean state $S^0$, ensuring that each perspective view is appropriately animated while maintaining global consistency. The underlying intuition is that if each perspective view undergoes its respective denoising process, the perspective video will feature meaningful animation. Moreover, if two perspective views overlap, they will align with each other (Jiménez, 2023; Bar-Tal et al., 2023; Lugmayr et al., 2022) to produce a seamless global animation.

Given a static input panorama $I$ and an initial spherical latent code $S^0 : \mathbb{S}^2 \to \mathbb{R}^{L \times c}$, we progressively remove noise employing a project-and-fuse procedure at each denoising step. Specifically, the spherical latent code at the $t^{\text{th}}$ denoising step, $S^t : \mathbb{S}^2 \to \mathbb{R}^{L \times c}$, is projected into multiple perspective latent codes $\mathcal{Z}^t = \{z_1^t, z_2^t, \ldots, z_n^t\}$, where each $z_k^t = \gamma(S^t, \boldsymbol{d}_k) \in \mathbb{R}^{p_h \times p_w \times (L \times c)}$ represents the $k^{\text{th}}$ perspective latent code projected in the equirectangular format detailed in Sec. 3.1. Each perspective latent code is then denoised by one step using a pre-trained perspective denoiser, denoted as $z_k^{t-1} = \Phi(z_k^t, i_k)$, where $i_k = \gamma(I, \boldsymbol{d}_k) \in \mathbb{R}^{p_H \times p_W \times c}$ is the perspective conditioning image projected from the panorama $I$. Subsequently, we optimize the spherical latent code $S^{t-1} : \mathbb{S}^2 \to \mathbb{R}^{L \times c}$ at step $t-1$ by fusing all the denoised perspective latent codes $z_k^{t-1}$. Formally, the denoising procedure at step $t$, denoted as $S^{t-1} = \Psi(S^t, I)$, encompasses the following operations:

$$\Psi\left(\mathcal{S}^t, I\right) = \operatorname*{argmin}_{\mathcal{S}} \mathbb{E}_{\boldsymbol{d} \in \mathbb{S}^2} \| \gamma(\mathcal{S}, \boldsymbol{d}) - \Phi\left(\gamma(\mathcal{S}^t, \boldsymbol{d}), \gamma(I, \boldsymbol{d})\right) \|. \tag{3}$$

## 3.4 Dynamic Panoramic Lifting

We define the panoramic video as $V = \{I^1, I^2, \ldots, I^L\}$, consisting of $L$ frames. The video is divided into overlapping perspective videos $\{v_0, v_1, \ldots, v_n\}$, each captured from specific camera directions $\{\boldsymbol{d}_1, \ldots, \boldsymbol{d}_n\}$, collectively encompassing the entire span of the panoramic video $V$. Subsequently, we estimate the geometry of the 4D scene by fusing the depth maps through spatial-temporal geometry alignment. Following this, we describe our methodology for 4D representation and the subsequent rendering procedure.

**Supervision from Spatial-Temporal Geometry Alignment.** To transition from 2D video to 3D space, we utilize a monocular depth estimator (Ranftl et al., 2021), inspired by advancements in (Zhou et al., 2024b), to estimate the scene's geometric structure. Nonetheless, depth maps generated for each frame and perspective might lack spatial and temporal consistency. To address this, we implement Spatial-Temporal Geometry Alignment using a pre-trained depth estimator $\Theta : \mathbb{R}^{h \times w \times 3} \to \mathbb{R}^{h \times w}$, applied to perspective images. Our objective is to amalgamate $n$ perspective depth maps $D_i^K = \Theta(\gamma(I^k, \boldsymbol{d}_i))$ into a cohesive panoramic depth map $D^k$ for each frame $I^k$, ensuring spatial and temporal continuity. We express these depth maps as a spherical representation $\mathcal{S}_D^1, \ldots, \mathcal{S}_D^L$. For enhanced optimization, we assign $n$ scale factors $\alpha_i^k \in \mathbb{R}$ and shifting parameters $\beta_i^k \in \mathbb{R}^{h \times w}$ to each perspective depth map. The comprehensive depth map $D^k$ is then optimized jointly with these parameters $\alpha$ and $\beta$. The formal objective is structured as follows:

$$\mathcal{S}_D^k = \operatorname*{argmin}_{\mathcal{S}} \mathop{\mathbb{E}}_{i \in \{1, \ldots n\}} \lambda_{\text{depth}} \mathcal{L}_{\text{depth}} + \lambda_{\text{scale}} \mathcal{L}_{\text{scale}} + \lambda_{\text{shift}} \mathcal{L}_{\text{shift}}. \tag{4}$$

where $\mathcal{L}_{\text{depth}} = \|\operatorname{softplus}(\alpha_i^k)\Theta(\gamma(I^k, d_i)) - \gamma(\mathcal{S}) + \beta_i^k\|$ is the depth supervision term, $\mathcal{L}_{\text{scale}} = \|\alpha_i^k - \alpha_i^{k-1}\| + \|\operatorname{softplus}(\alpha_i^k) - 1\|$ the regularize term for $\alpha$, and $\mathcal{L}_{\text{shift}} = \mathcal{L}_{\text{TV}}(\beta_i^k) + \|\beta_i^k - \beta_i^{K-1}\|$ the regularize term for $\beta$ where $\mathcal{L}_{\text{TV}}$ is the TV regularization.

**4D Representation and Rendering.** We represent and render the dynamic scene using $T$ sets of 3D Gaussians. Each set, corresponding to a specific timestamp $t$, is denoted as $G_t = \{(\boldsymbol{p}_t^i, \boldsymbol{q}_t^i, \boldsymbol{s}_t^i, \boldsymbol{c}_t^i, o_t^i) | i = 1, \ldots, n\}$. This definition aligns with the methods described in (Bahmani et al., 2024), which also provides a fast rasterizer for rendering images based on these Gaussian sets and given camera parameters. Consistent with Sec. 3.1, while the camera intrinsics remain fixed, we parameterize the camera extrinsics through a position $\boldsymbol{p} \in \mathbb{R}^3$ and an orientation $\boldsymbol{d} \in \mathbb{S}^2$. The training process is structured in two stages: initially, we directly supervise the 3D Gaussians using the panoramic videos. Let $\mathcal{R}(G, \boldsymbol{p}, \boldsymbol{d})$ represent the rasterized image from Gaussian set $G$, utilizing camera extrinsics $\boldsymbol{p} = 0$ and camera direction $\boldsymbol{d}$. Let $I_t$ denote the $t^{\text{th}}$ frame of the panoramic video. We optimize the $t^{\text{th}}$ Gaussian set $G_t$ using the following objective:

$$\mathcal{L} = \lambda_{\text{rgb}} \mathcal{L}_{\text{rgb}} + \lambda_{\text{temporal}} \mathcal{L}_{\text{temporal}} + \lambda_{\text{sem}} \mathcal{L}_{\text{sem}} + \lambda_{\text{geo}} \mathcal{L}_{\text{geo}} \tag{5}$$

where the RGB supervision term $\mathcal{L}_{\text{rgb}} = \lambda \mathcal{L}_1 + (1 - \lambda)\mathcal{L}_{\text{SSIM}}$ is the same as 3D-GS (Kerbl et al., 2023), and the temporal regularize term $\mathcal{L}_{\text{temporal}}$ written as:

$$\mathcal{L}_{\text{temporal}} = \sum_{i=1}^{n} \|\mathcal{R}(G_t, \boldsymbol{0}, \boldsymbol{d}_i) - \mathcal{R}(G_{t-1}, \boldsymbol{0}, \boldsymbol{d}_i))\| \tag{6}$$

Then, we adopt the distillation loss and geometric regularization used in (Zhou et al., 2024b), the distillation loss is defined as follows: $\mathcal{L}_{sem} = 1 - \cos \langle \text{CLS}(\mathcal{R}(G_t, \boldsymbol{0}, \boldsymbol{d}_i)), \text{CLS}(\mathcal{R}(G_t, \boldsymbol{\delta}_p, \boldsymbol{d}_i)) \rangle$, where $\boldsymbol{\delta}_p \in [-\alpha, \alpha]^3$ is the disturbing vector, $\text{CLS}(\cdot)$ the feature extractor such as DINO (Oquab et al., 2023), and $\cos\langle \cdot, \cdot \rangle$ the cos value of two vectors. The geometric regularization is defined as follows: $\mathcal{L}_{geo} = 1 - \frac{\text{Cov}(\mathcal{R}_D(G_t, \boldsymbol{0}, \boldsymbol{d}_i), \Theta(\gamma(I, \boldsymbol{d}_i)))}{\sqrt{\text{Var}(\mathcal{R}_D(G_t, \boldsymbol{0}, \boldsymbol{d}_i)) \text{Var}(\Theta(\gamma(I, \boldsymbol{d}_i)))}}$, where $\mathcal{R}_D$ is the rendered depth, $\text{Cov}(\cdot, \cdot)$ the covariance, and $\text{Var}(\cdot)$ the variance.

# 4 EXPERIMENTS

## 4.1 EXPERIMENTAL SETTINGS

**Implementation Details.** For perspective images, we uniformly select 20 directions $\boldsymbol{u}$ on the sphere $\mathbb{S}^2$ as the z-axis of 20 cameras. In each experiment, the image plane size $\boldsymbol{s}$ is set at $0.6 \times 0.6$, with a focal length $f = 0.6$ and a resolution of $512 \times 512$. Rotation along the z-axis is kept at zero for all cameras, ensuring that the up-axis for the $i^{\text{th}}$ camera aligns with the $(O, \boldsymbol{u}_i, \boldsymbol{z})$ plane. During the animating phase, we utilize the perspective denoiser $\Phi$, instantiated as the Animate-anything model (Dai et al., 2023), which fine-tunes the SVD model (Blattmann et al., 2023a). In the Spatial-Temporal Geometric Alignment stage of the lifting phase, the depth estimator $\Theta$ is implemented using MiDaS (Ranftl et al., 2021; Birkl et al., 2023). All experiments are executed on a single NVIDIA A100 GPU with 80 GB RAM.

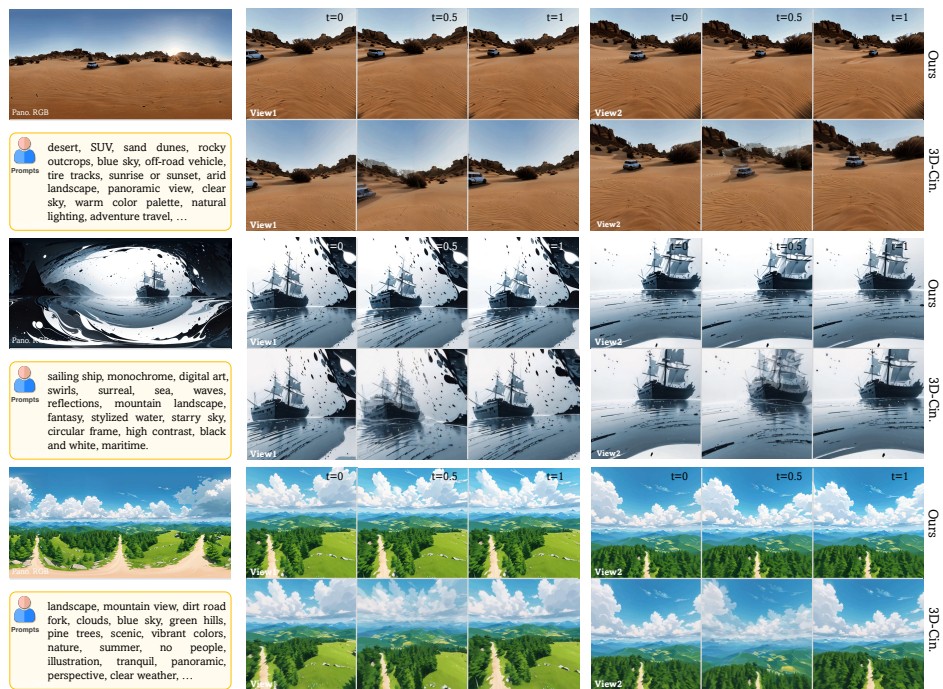

Figure 4: **Comparison between 4K4DGen and 3D-Cinemagraphy.** We present the input static panorama (Pano RGB), the corresponding text prompts, and the rendered results from different views and at various timestamps. 4K4DGen (Ours) effectively generates 4D scenes that are both spatially and temporally consistent, while 3D-Cinemagraphy (3D-Cin.) suffers from ghosting artifacts in the middle frames.

**Evaluation.** As there is no ground truth 4D scene data available, we render videos at specific test camera poses from the synthesized 4D representation and employ non-reference video/image quality assessment methods for quantitative evaluation of our approach. For the test views, we select random cameras with $p = 0$ as part of our testing camera set. We then introduce disturbances as described in Sec. 3.4, applying a disturbance factor of $\alpha = 0.05$ at these selected views. Datasets. The task of generating 4D panoramas from static panoramas is new, and thus, no pre-existing datasets are available. In line with previous large-scale scene generation works (Zhou et al., 2024b; Yu et al., 2024), we evaluate our methodology using a dataset of 16 panoramas generated by text-to-panorama diffusion models (Yang et al., 2024). Baselines. Current SDS-based methods (Wu et al., 2024a; Zhao et al., 2023) are limited to generating object-centered assets and do not support outward-facing scene generation. We compare our method with the optical-flow-based 3D dynamic image technique, 3D-Cinemagraphy (3D-Cin.) (Li et al., 2023b) (both the "circle" and "zoom-in" mode), by inputting the static panorama and projecting the output onto perspective images. Metrics. It is challenging to evaluate the visual quality without a ground-truth reference. We evaluate the rendered perspective videos regarding both the frame and video visual quality. For frame quality, We employ the LLM-based visual scorer Q-Align (Wu et al., 2023a) (IQ Scorer and IA Scorer) to evaluate the quality of individual frames. For video quality, we use the Q-Align video model (VQ) as the quality scorer. Additionally, we conduct user studies to further evaluate the results. In this paper, there are two types of user studies: (1) User Choice (UC), where participants are asked to compare and select the best video from candidates generated by different methods, and (2) User Agreement (UA), where participants assess whether specific properties are present in the videos generated by a particular approach.

## 4.2 RESULTS

**Quantitative Results.** We show the quantitative comparison between 4K4DGen and 3D-Cinemagraphy (Li et al., 2023a) in Tab. 1. 4K4DGen consistently achieves better performance in the LLM-based Q-Align metric regarding the image quality (IQ), image aesthetic (IA), and the

Table 1: **Comparison with 3D-Cinemagraphy.** We compare our method with 3D-Cinemagraphy using rendered images from 4D representations. The IQ, IA, and VQ models represent the image quality scorer, image aesthetic scorer, and video quality scorer, respectively, within the Q-Align assessment framework. Our method, 4K4DGen, consistently achieves superior performance in both image and video quality across these metrics. Furthermore, 4K4DGen performs better in our user studies in terms of visual quality (Quality), motion amplitude (Amplitude), and the motion naturalness (Naturalness). Please refer to D.2 for further details.

| Method | Q-Align (IQ) ↑ | Q-Align (IA) ↑ | Q-Align (VQ) ↑ | Quality (UC) ↑ | Amplitude (UC) ↑ | Naturalness (UC) ↑ |
|---|---|---|---|---|---|---|
| 3D-Cinemagraphy (zoom-in) | 0.47 | 0.38 | 0.57 | 7% | 29.4% | 19.7% |
| 3D-Cinemagraphy (circle) | 0.48 | 0.40 | 0.58 | 12% | 32.0% | 21.1% |
| Ours (holistic pipeline) | **0.66** | **0.44** | **0.62** | **81%** | **38.6%** | **59.2%** |

video quality (VQ). Besides, 4K4DGen is preferred by the users considering the video quality, motion amplitude, and motion naturalness.

**Qualitative Results.** We present a qualitative comparison between 4K4DGen and 3D-Cinemagraphy (3D-Cin.) on the rendered images from 4D representations. Since the performance of 3D-Cin. is similar under the "circle" and "zoom-in" settings in Tab. 1, we use the "circle" setting to represent 3D-Cin. in Fig. 4. As shown in the figure, 4K4DGen produces high-quality perspective videos that maintain consistency across both time and views, whereas 3D-Cin. struggles with generating ghosting artifacts in the middle frames.

### 4.3 ABLATION STUDIES

We conduct ablation studies for both the animating and lifting phases of our methodology. In the animating phase, we perform evaluation on 2D animated videos with different strategies, and highlight the importance of our spherical denoise strategy by replacing it with two basic animation techniques. In the lifting phase, we analyze the impact of excluding the Spatial-Temporal Geometry Alignment process and the temporal loss during the optimization of 4D representations.

**Animating Phase.** For analyzing the strategies in the animating phase, as shown in Tab. 2, we use Q-Align (visual quality scorer), view-consistency (user agreement), motion amplitude (user choice), and motion naturalness (user choice) to evaluate the 2D animated videos. For the details of the user studies, please refer to the Appendix D.2. To animate the panorama into a panoramic video, a straightforward approach is to apply animators directly to the entire panorama. However, we observed that this strategy often results in minor motion, as shown in Fig. 5 (b) and Tab. 2 (Animate Pano. with small motion amplitude and less naturalness). This issue arises due to two main reasons: (1) animators are typically trained on perspective images with a narrow field of view (FoV), whereas panoramas have a 360° FoV with specific distortions under the equirectangular projection; (2) our panorama is high-resolution (4K), which exceeds the training distribution of most 2D animators and can easily cause out-of-memory issues, even with an 80GB VRAM graphics card. Thus the

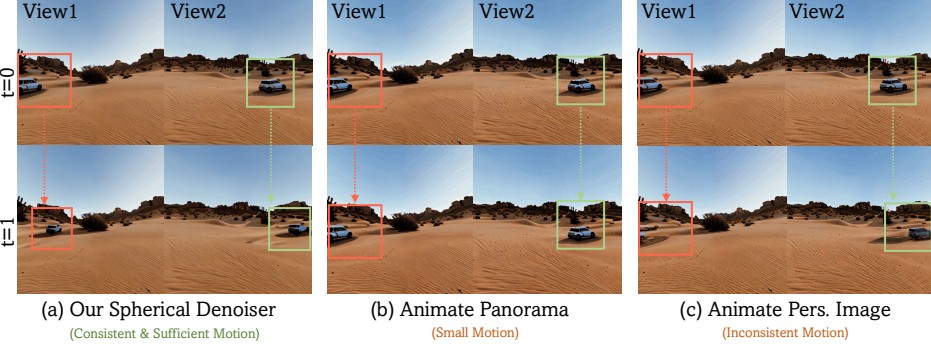

(a) Our Spherical Denoiser
(Consistent & Sufficient Motion)

(b) Animate Panorama
(Small Motion)

(c) Animate Pers. Image
(Inconsistent Motion)

Figure 5: **Comparison to Different Animators**: Animators trained primarily on perspective images tend to produce limited motion when applied to panoramas, and the resolution may be limited. On the other hand, animating perspective images individually can lead to inconsistencies between overlapping views.

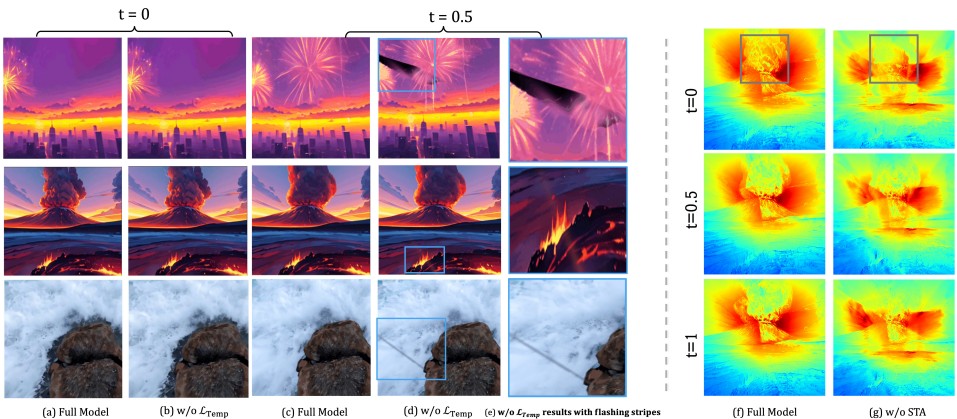

Figure 6: **Ablating Lifting Phase**: (Left) The w/o $\mathcal{L}_{\text{Temp}}$ variant (column d) produces renderings with flashing stripes. Zoomed-in details of the flashing stripe region are highlighted in (e). (Right) Without spatial-temporal geometry alignment, the geometry in the smoke area of the volcano for the w/o STA variant (column g) appears less consistent compared to the full model (column f).

panoramas have to be down-sampled to a lower resolution (2K), causing a loss of details. To this end, we seek to animate on perspective views. Applying the animator on perspective views offers benefits such as reduced distortion and inputs that suit the domain of the animator, allowing for smooth animation of high-resolution panoramas. However, animating perspective images separately can introduce inconsistencies between overlapping perspective views, as illustrated in Fig. 5 (c) and Tab. 2 (Animate Pers.). To resolve this challenge, we propose simultaneously denoising all perspective views and fusing them at each denoising step, in the spherical latent space, which capitalizes on the benefits of animating perspective views while ensuring cross-view consistency. The results are displayed in Fig. 5 (a) and Tab. 2.

**Lifting Phase.** We conduct ablation studies on the Spatial-Temporal Geometry Alignment (STA) module and the temporal loss during the lifting phase, as shown in Fig. 6.

Table 2: **Different Animation Strategies in the Animating Phase**. We analyze the efficacy of animation strategies by evaluating the animated 2D videos in different ways. Animating the entire panorama results in worse motion and reduced resolution (first row), as indicated by the Amplitude and Naturalness metric. Conversely, animating from perspective views leads to inconsistencies across different views (second row), as supported by the Q-Align metric and the "View-consistency (UA)" study. 4K4DGen capitalizes the generative ability from perspective animating priors while enabling cross-view consistent motion between different perspectives, which achieves the best motion naturalness and amplitudes among all the settings (third row).

| Animator | Max Pano. Res. | Q-Align (VQ) ↑ | View-consistency (UA) ↑ | Amplitude (UC) ↑ | Naturalness (UC) ↑ |
|---|---|---|---|---|---|
| Animate Pano. | $2048 \times 1024$ | 0.82 | - | 26.8% | 17.8% |
| Animate Pers. | $4096 \times 2048$ | 0.64 | 33% | 32.4% | 39.3% |
| Ours (Animating Phase) | $4096 \times 2048$ | **0.85** | **70%** | **40.8%** | **42.9%** |

## 5 CONCLUSION

**Conclusion.** We have proposed a novel framework **4K4DGen**, allowing users to create high-quality 4K panoramic 4D content using text prompts, which delivers immersive virtual touring experiences. To achieve panorama-to-4D even without high-quality 4D training data, we integrate generic 2D prior models into the panoramic domain. Our approach involves a two-stage pipeline: initially generating panoramic videos using a Panoramic Denoiser, followed by 4D elevation through a Spatial-Temporal Geometry Alignment mechanism to ensure spatial coherence and temporal continuity.

**Limitation.** First, the quality of temporal animation in the generated 4D environment mainly relies on the ability of the pre-trained I2V model. Future improvements could include the integration of a more advanced 2D animator. Second, since our method ensures spatial and temporal continuity during the 4D elevation phase, it is currently unable to synthesize significant changes in the environment, such as the appearance of glowing fireflies or changing weather conditions. Third, the high-resolution and time-dependent representation of the generated 4D environment necessitates substantial storage capacity, which could be optimized in future work using techniques such as model distillation and pruning.

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

## A  APPENDIX

Due to space constraints in the main draft, we include supplementary details and experimental results in the appendix. Specifically, in Sec. B , we provide details about the acquisition process for the static panoramas used in our experiments. In Sec. C, we offer further explanation of the implementation for both the animation and lifting phases. Finally, in Sec. D, we describe the experimental setup and present additional results.

## B  ACQUISITION OF PANORAMAS

The static panoramas used in the dataset of the main draft are generated by a text-to-panorama diffusion model, fine-tuned from stable diffusion  (Rombach et al., 2022) on SUN360. Similar to  (Yang et al., 2024; Feng et al., 2023), this model follows three steps: circular blending, super-resolution, and refinement. The panoramas are initially at a resolution of $6144 \times 3072$ and then down-sampled to $4096 \times 2048$ using the bi-linear interpolation.

## C  IMPLEMENTATION DETAILS

In this section, we introduce the implementation details of the panoramic animator and the 4D lifting procedure.

**Implementation of Spherical Representing**   For the spherical representation, the continuous spherical mapping $\mathcal{S}_I : \mathbb{S}^2 \to \mathbb{R}^C$ is instantiate as discrete point set $\mathcal{P} = \{p_i\}$, which is uniformly sampled from the sphere $\mathcal{S}_I$. We first initialize a icosahedron with 20 triangle faces $\{f_i | i = 1, \cdots, 20\}$ to approximate a real sphere $\mathbb{S}^2$. Then we uniformly sample a point set $P_i$ on each face $f_i$ and union all the point sets together as $\hat{\mathcal{P}} = \cup_{i=1}^{20} P_i$. We then obtain the discrete point set $\mathcal{P}$ by projecting $\hat{\mathcal{P}}$ onto the sphere $\mathbb{S}^2$ by $\mathcal{P} = \{p_i/\|p_i\| \mid p_i \in \hat{\mathcal{P}}\}$.

**Panoramic Animation Phase**   For the Panoramic Animator, we set the video length $L = 14$, the channel number $c = 9$, the latent code size $(h, w) = \frac{1}{8}(H, W)$, the perspective image size $p_H = p_W = \frac{1}{4}W$. The sphere is uniformly divided into 20 perspective views, each with $80°$ FOV. For the denoiser, the max denoising step is 25. For the continuous optimization in Eq. 3, we calculate each latent vector at each point on the sphere by taking the weighted average on the latent vectors of knn points that are projected from the corresponding pixel on the perspective views, the weights are the inversed distances between the sphere point and the projected points. We conduct the spherical denoising for the first 10 denoising steps and then the spherical latent is projected to the equirectangular form and denoised using sliding window, to avoid noises introduced by interpolation. The perspective denoiser is initiated as Animate-Anything  (Dai et al., 2023). The masks required by the denoiser are given by bounding boxes defined by user clicks.

**Dynamic Panoramic Lifting Phase**   In the lifting phase, similar to the animation phase, we choose the perspective view number $n = 20$, each with $80°$ FOV. Each perspective view has a square shape, $P_H = P_W = \frac{1}{4}W$, where $W$ is the width of the original static panorama. In the Spatial-Temporal Geometric Alignment stage, the depth estimator $\Theta$ is implemented using MiDaS  (Ranftl et al., 2021; Birkl et al., 2023). The depth map from the perspective image is scaled according to the projection of the unit-length ray direction onto the camera orientation $\boldsymbol{d}$. Formally, if the pre-scaled depth is $d$ at point $p \in \hat{\mathcal{P}}$ introduced above, the scaled depth should be $d/\|p\|$. Additionally, for scenes without distinct boundaries, such as the sky, depth values for distant elements are assigned a finite value to support optimization.

**Optimization Details**   The hyper-parameters for optimization are set as follows: $\lambda_{\text{depth}} = 1, \lambda_{\text{scale}} = 0.1, \lambda_{\text{shift}} = 0.01$. We conduct Spatial-Temporal Geometry Alignment optimization over 3000 iterations, with $\lambda_{\text{scale}}$ and $\lambda_{\text{shift}}$ set to zero during the first 1500 iterations. For the 4D representation training stage, Gaussian parameters are optimized over 10000 iterations for each time stamp $t$. The hyper-parameters for this stage are defined as $\lambda_{\text{rgb}} = 1, \lambda_{\text{temporal}} = \lambda_{\text{sem}} = \lambda_{\text{geo}} = 0.05$,

and the disturbance vector range $\alpha$ is varied at $0.05$, $0.1$, and $0.2$ during the $5400$, $6600$, and $9000$ iterations, respectively.

# D   EXPERIMENTAL DETAILS

## D.1   THE PROCEEDING TIME OF PER-GENERATION

**Animating Phase.**   We provide the time and GPU cost to animate a single video at different resolutions in the Table 3.

Table 3: **The Proceeding Time of Animating Phase.**

| Resolution | GPU Usage (GB) ↓ | Time Cost (Minutes/frame) ↓ |
|---|---|---|
| $1024 \times 512$ (1K) | 9.38 | 0.89 |
| $2048 \times 1024$ (2K) | 29.31 | 2.99 |
| $4096 \times 2048$ (4K) | 73.41 | 22.31 |

**Lifting Phase.**   We provide the time and GPU cost to lift a single frame at 4K, 2K, and 1K resolution. The results are shown in the Table 4.

Table 4: **The Proceeding Time of Lifting Phase.**

| Resolution | GPU Usage (GB) ↓ | Time Cost (Minutes/frame) ↓ |
|---|---|---|
| $1024 \times 512$ (1K) | 7.59 | 19 |
| $2048 \times 1024$ (2K) | 12.07 | 22 |
| $4096 \times 2048$ (4K) | 31.27 | 33 |

The computational cost increases rapidly with the resolution, making 4K generation highly challenging. We will put more engineering effort to accelerate the generation pipeline in the future.

## D.2   USER STUDY DETAILS

### D.2.1   USER STUDY FOR VIDEO QUALITY

We conducted two user studies, gathering a total of 84 questionnaires from 42 users. For the "Quality (UC)"column in Tab. 1, we collected 42 questionnaires, each containing eight questions. Each question asked users to choose the bests video in term of visual quality from the perspective videos provided by different models. The user choice (UC) score of a method is the percentage of times the method's video was selected as the best one, out of a total of 336 questions. Thus, the UC scores for all methods sum to 100%. For the "View-Consistency (UA)" column in Tab. 2, we collected another 42 questionnaires, with each questionnaire containing eight questions. Each question presented two videos from different views, both generated by the same method, and users were asked to determine whether the two videos were view-consistent. The user agreement (UA) score is the percentage of video pairs marked as view-consistent out of all the video pairs generated by the method. The UA scores do not necessarily sum to 100%. In the Quality column of Tab. 1, among the 336 questions, users selected 4K4DGen 272 times, 3D-Cin. (circle) 40 times, and 3D-Cin. (zoom-in) 24 times. In the View-consistency column of Tab. 2, 118 out of 168 video pairs generated by "Our" were marked as consistent, while 56 out of 168 pairs from "Animate Pers" were considered consistent.

### D.2.2   USER STUDY FOR VIDEO MOTION

Since the quantitative evaluation of motion quality remains an open problem in our tasks, we hereby conducted supplemented user studies for the "4D generation task" and "Animating Phase", which considers the motion's naturalness and amplitude.

- Motion's naturalness: the motion of the generated view should be natural to human's understanding, avoiding abrupt pixel changes across frames.

- Motion's amplitude: the motion trajectory of the **scene's subjects** should have adequate and realistic magnitude.

The user studies of Amplitude column and the Naturalness column are all conducted in the "user choice" (UC) way. For each Amplitude and Naturalness column in Tab. 1 and Tab. 2, we collect 320 questions from 20 participants. Each questions contains three videos from the different three methods, users are asked to select the best one or more videos from the provided set that exhibit noticeably greater amplitude (for the Amplitude columns) or superior naturalness (for the Naturalness columns). Since users could select more than one video per question, the UC metric was normalized based on the total number of selections. For example, if method A, B, C are selected $n_a$, $n_b$, and $n_c$ times, respectively, the UC metric for them should be $\frac{n_a}{n_a+n_b+n_c}$, $\frac{n_b}{n_a+n_b+n_c}$, and $\frac{n_c}{n_a+n_b+n_c}$.

### D.3 MORE QUANTITATIVE RESULTS

We present quantitative results on an additional **32** scenes randomly sampled from WEB360 dataset (Wang et al., 2024b). As shown in the Table 5, 4K4DGen consistently outperforms the baseline methods across the quantitative metrics.

Table 5: **Comparison with 3D-Cinemagraphy in WEB360 Dataset.** We adopt the FVD (Unterthiner et al., 2019) and KVD (Unterthiner et al., 2018) to evaluate the generated panoramic video, which is the intermediate result from the animating phase. The IQ, IA, and VQ models represent the image quality scorer, image aesthetic scorer, and video quality scorer, respectively, within the Q-Align assessment framework.

| Method | FVD ↓ | KVD ↓ | Q-Align (IQ) ↑ | Q-Align (IA) ↑ | Q-Align (VQ) ↑ |
|---|---|---|---|---|---|
| 3D-Cinemagraphy (zoom-in) | 307 | 5.86 | 0.65 | 0.57 | 0.70 |
| 3D-Cinemagraphy (circle) | 309 | 5.72 | 0.65 | 0.57 | 0.70 |
| 4K4DGen | **218** | **1.76** | **0.73** | **0.64** | **0.77** |

### D.4 COMPARISONS WITH MORE BASELINE METHODS

To further address your primary concern, we adopt two types of baseline methods: 4D Object Generation and 4D Scene Generation, to compare with the proposed method. Note that the panoramic 4D generation is still underexplored due to the scarcity of annotated data and the lack of well-trained prior models tailored for panorama format. As a result, we find that existing methods cannot achieve similar quality as 4K4DGen in this task.

**4D Object Generation methods.** Following your suggestion, we have devoted substantial engineering efforts to adapt the 4D object generation framework 4DGen (Yin et al., 2023) to our camera settings, as shown in the qualitative results in Figure 8 of the revised paper. It demonstrates that recent 4D object generation methods struggle to generate scene-level content due to inherent domain gaps between objects and scenes. Our method overwhelmingly outperforms 4DGen in terms of quantitative evaluations (e.g., Image Quality, Image Aesthetics, and Video Quality) and qualitative evaluations.

**4D Generation Methods for Scene.** We also construct two 4D scene generation baselines: (1) We equip LucidDreamer (Liu et al., 2023) with our animator. We follow the authors' setting, using ZoeDepth and its inpainting model (Stable Diffusion) to expand invisible views for each timestamp, and then we use our backbone animator to animate and optimize the Gaussians. (2) We employed a very recent 4D scene generation technique DimensionX (Sun et al., 2024). We use the default configuration, employing its lora ("orbit left") model to generate novel views and 3D structures. Since the DimensionX's T-Director is currently unavailable, we leveraged the same backbone animator from our approach to provide temporal guidance for its 4D representation in the 4D generation stage. Compared with existing methods, including 4DGen, LucidDreamer, and DimensionX, our method consistently achieves higher quantitative results, demonstrating the efficacy of the proposed method.

Table 6: **Comparison with 4D Generation Methods.**

| Method | Q-Align (IQ) ↑ | Q-Align (IA) ↑ | Q-Align (VQ) ↑ |
|---|---|---|---|
| 4DGen (object) | 0.19 | 0.20 | 0.29 |
| 3D-Cinemagraphy (zoom-in) | 0.47 | 0.38 | 0.57 |
| 3D-Cinemagraphy (circle) | 0.48 | 0.40 | 0.58 |
| LucidDreamer + Our Animator | 0.44 | 0.41 | 0.58 |
| DimensionX + Our Animator | 0.55 | 0.42 | 0.60 |
| 4K4DGen | **0.66** | **0.44** | **0.62** |

### D.5 MORE QUALITATIVE RESULTS

We provide additional qualitative results in Figure 7. Furthermore, we highly recommend viewing the video renderings of 4K4DGen and comparisons to baseline models in the supplementary static HTML page for a more comprehensive and immersive experience.

We adapt the 4D object generation framework 4DGen (Yin et al., 2023) to our specific settings and present the qualitative results in Figure 8, which indicate that the generated object varies significantly in form from 4K4DGen's scene outputs. We also compare OmniNeRF (Gu et al., 2022)'s optimized geometry with 4K4DGen. The corresponding depth maps are shown in Figure 9. It can be evidently demonstrated that 4K4DGen attains sharper geometric results. We provide the renderings of a lifted 3D scene where a user walked along a street in Figure 10. Notice the roof highlighted by green bounding boxes in (a) and (b). When the user walks nearer and more area of the roof is observed, it implies the necessity of the lifted 3D structure.

## E ETHICS AND REPRODUCIBILITY STATEMENT

**Ethics Statement.** Our research enables the generation of 4D digital scenes from a single panoramic image, which is advantageous for various applications such as AR/VR, movie production, and video games. This technology distinctly excels in creating high-resolution 4D scenes up to 4K, significantly enhancing user experiences. However, there is potential for misuse in the creation of deceptive content or privacy violations, which contradicts our ethical intentions. These risks can be mitigated through a combination of regulatory and technical strategies, such as watermarking.

**Reproducibility.** We provide sufficient implementation details to reproduce our methodology in Sec. C, including the details of spherical denoiser, panoramic animator, dynamic panoramic lifting, etc. We provide 16 Sec. 4's panoramas and Sec. D.4's 32 panoramas in the revised supplementary material. Furthermore, we will make our panorama datasets and related code publicly available in the future.

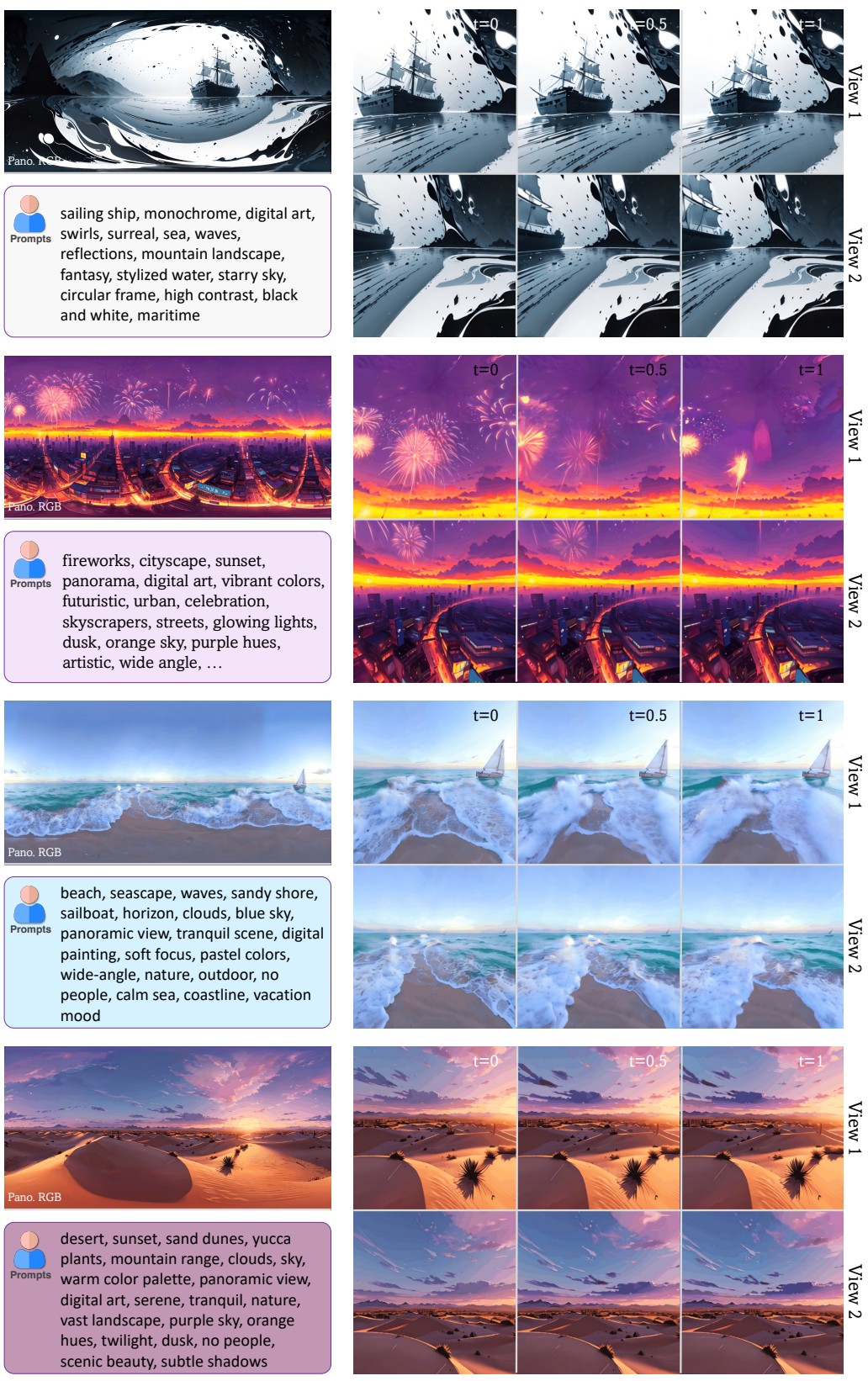

Figure 7: **Visualizations**: We provide more visual results. For each shown case we provide the input panorama, corresponding text prompt, and the rendering from two perspective views.

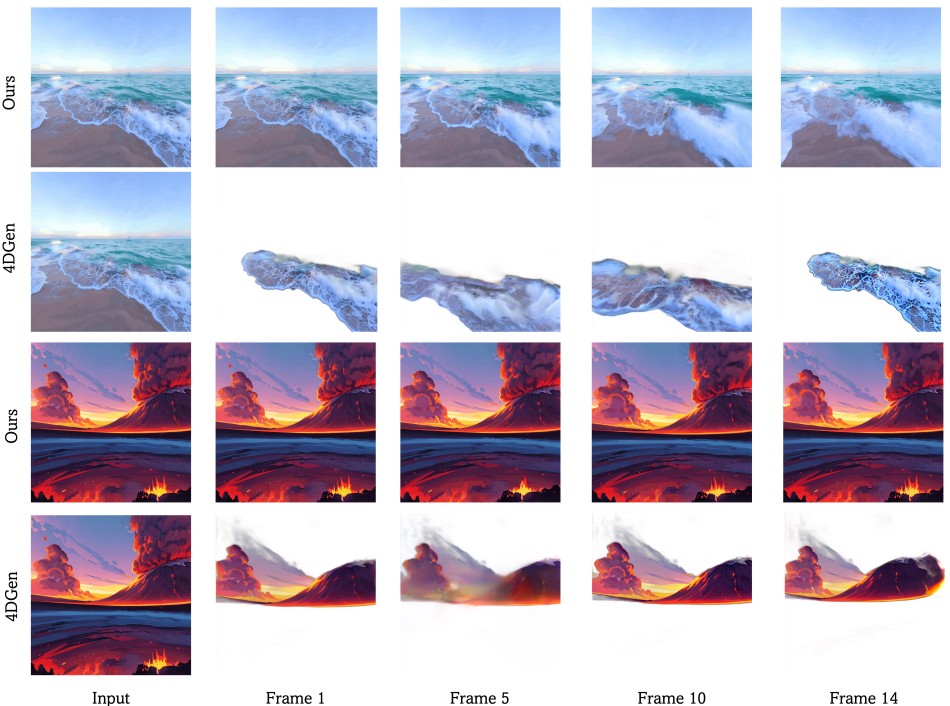

Figure 8: **Results of 4DGen**: 4DGen (Yin et al., 2023) focuses on object generation and struggles to generate scenes.

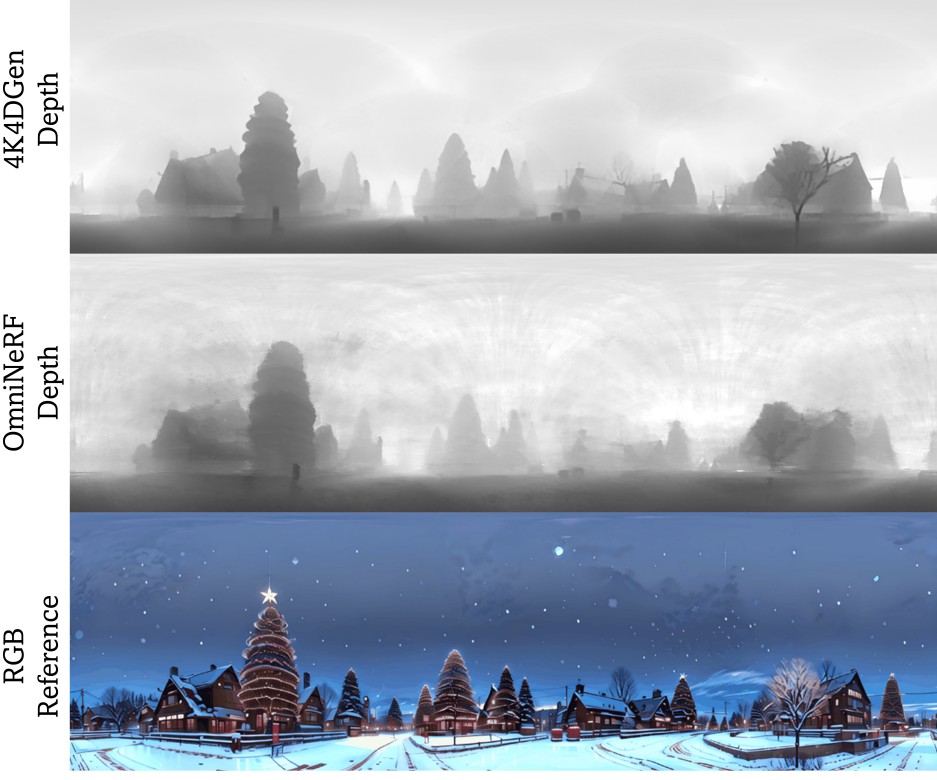

Figure 9: **Results of OmniNeRF**: The optimized geometry of OmniNeRF (Gu et al., 2022) is not as sharp as 4K4DGen.

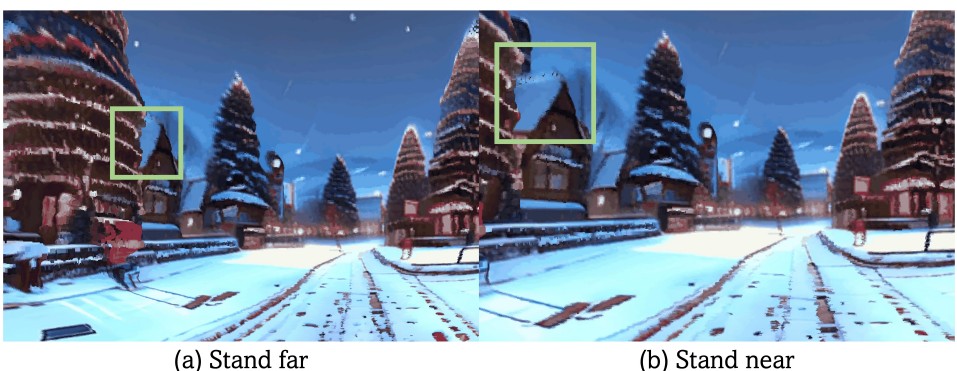

(a) Stand far              (b) Stand near

Figure 10: **Occlusion in True 3D Structure**: When the user walks nearer, the more area of the roof (highlighted by the green box) will be observed. It is hard to implement such effect without the lifted 3D structure.

