# OpenReview forum: "4K4DGen: Panoramic 4D Generation at 4K Resolution"
_ICLR.cc/2025/Conference — ICLR 2025 Spotlight_

### Official Review · Reviewer_jzFB · 2024-11-02

**Soundness:** 3
**Presentation:** 2
**Contribution:** 3
**Rating:** 8
**Confidence:** 5

**Summary:**

The paper first proposes a method, 4D4KGen, which can generate immersive 360-degree, 4K panoramic 4D videos from a single 4K static panoramic image. The method first uses the pre-trained perspective video generative model AnimateAnything to create a panoramic video through a project-and-fuse scheme with a spherical latent space. Next, the pretrained monocular depth estimator MiDaS is used to create a panoramic depth map with a spherical representation. Finally, the 4D animated panoramic videos are rendered using reconstructed 3DGS. Qualitative results show that the proposed method can effectively generate immersive 360-degree 4D panoramic videos.

**Strengths:**

(1) The paper first proposes animatable 360-degree 4K panoramic images.

(2) To create 360-degree panorama with fixed-size pretrained video generation and depth estimation models, the authors carefully design the projection function.

(3) The qualtative results shows that great spatial and temporal consistency.

**Weaknesses:**

(1) I understand that there is no metric to evaluate 360-degree panoramic images, but there is insufficient basis for measuring FID and KID between the generated frames and the corresponding perspective images projected from the static panoramas. In particular, these metrics may be meaningless when the reference set is small, and in this paper, only 16 panoramic videos were generated. Proposing and measuring such less meaningful metrics, especially when this paper aims to serve as a baseline for future 4D 360-degree panorama video generation, does not seem beneficial for subsequent research. Additionally, if we want to assess the fidelity of the generated video, we could measure FVD [1] and KVD [1]. Therefore, I recommend removing FID and KID.

(2) I don't know how long the proposed method takes, but generating only 16 videos seems too few. To demonstrate that the proposed method works generally on many 4K 360-degree panoramic images, it would be better to generate more videos (at least 30 or more) to measure quantitative results.

(3) In Sec. 4.3 Ablation Studies, a straightforward approach would be to apply animators directly to the entire panorama as the direct baseline, measured by the Q-Align metrics (IQ, IA, and VQ) and also through a user study. If 2D animators cause out-of-memory issues, we can simply upsample each frame of the videos using a super-resolution (upsampling) model such as StableSR [2].

(4) In Fig. 6, Ablation of the Lifting Phase, I find it difficult to visually discern the difference between the proposed temporal regularization and Spatial-Temporal Geometry Alignment (STA). The authors should have included a video of this in the supplementary materials or shown a more definitive difference in the figure.

(5) Since the paper's topic is closely related to panoramic image generation, the following papers [3, 4, 5] should be mentioned in the related work section.

(6) Please double-check the citations (bibtex) to confirm which papers were accepted by specific conferences. Too many papers published in conferences are listed as arXiv.


The proposed method demonstrates good quality in the new research field of 4D panoramic video generation, but it lacks convincing evidence for the choice of pipeline components and is missing many essential elements that a paper should fundamentally include. I believe it lacks the basic qualities necessary for publication in a top-tier ML conference ICLR.


[1] Unterthiner et al., Towards Accurate Generative Models of Video: A New Metric & Challenges, arXiv preprint arXiv:1812.01717, 2018.

[2] Wang et al., Exploiting Diffusion Prior for Real-World Image Super-Resolution, IJCV2024

[3] Bar-Tal et al., MultiDiffusion: Fusing Diffusion Paths for Controlled Image Generation, ICML 2023

[4] Lee et al., SyncDiffusion: Coherent Montage via Synchronized Joint Diffusions, NeurIPS 2023

[5] Quattrini et al., Merging and Splitting Diffusion Paths for Semantically Coherent Panoramas, ECCV 2024

**Questions:**

(1) How much time and GPU memory are required to generate a single 4K video?

---

> ### Author Response · Authors · 2024-11-22
> **Response to Reviewer jzFB - Part1**
>
> **Clearance**
>
> We want to clarify that 4K4DGen aims to generate panoramic 4D scenes in dynamic Gaussian Splatting, which allows dynamic novel view rendering with stereoscopic effect and delivers immersive experiences in VR/XR applications. The holistic pipeline requires considerate designs to achieve both spatial and temporal consistency in 4K resolution, which is beyond the 2D panoramic video generation task. Since the evaluation of 4D scene generation is an open problem due to the lack of ground truth data, we evaluate our approach by measuring the visual quality of perspective renderings.
>
> **W1: FID/KID are not suitable metrics for evaluating the video generation task.**
>
> Thank you for the feedback. Following your suggestion, we have adopted the FVD [1] and KVD [1] to evaluate the generated panoramic video, which is the intermediate result from the animating phase. Specifically, we conducted the evaluation on a random-chosen subset of WEB360 [2], which contains 32 panoramas. The results are shown below. We will scale up the test set to **128**  in the discussion period.
>
>  | Method  |   FVD  $\downarrow$  | KVD $\downarrow$   |   IQ $\uparrow$  |   IA $\uparrow$    |     VQ   $\uparrow$  |
>  | ------------- |:-------------:|:-------------:|:-------------:|:-------------:|:-------------:|
>  | 3D-Cinemagraphy (zoom-in) |        307          |     5.86   |    0.65    |    0.57    | 0.70   |
>  | 3D-Cinemagraphy (circle)     |        309          |        5.72     | 0.65    |   0.57   | 0.70 |
>  | 4K4DGen                               |        218         |        1.76     |     0.73         |  0.64 |     0.77    |
>
> **W2: The test set is small (e.g., 16 scenes) to measure quantitative results?**
>
> As mentioned above, we have presented results on an additional 32 scenes sampled from WEB360 [2]. As shown in the table,  4K4DGen consistently outperforms the baseline methods across the metrics . Further results will be released during the discussion phase.
>
> **W3 Evaluating the straight-forward baseline, in which the entire panorama is animated by the off-the-shelf animator?**
>
> We evaluated a baseline approach where the animator is directly applied to the entire panorama at 2K resolution, as shown in the first row of Table 2 (Animate Pano.). We also tested the results at 4K resolution using a super-resolution technique [3], but the improvements were not significant.
>
> Since the animator is trained on low-resolution data, directly using the straight-forward baseline would result in limited motion as highlighted in the fourth column of Table 2 and illustrated in Fig. 5(b). Super-resolution will not address the problem of limited motion if the original low-resolution video has a small motion.

---

> ### Author Response · Authors · 2024-11-22
> **Response to Reviewer jzFB - Part2**
>
> **W4: Difficult to visually discern the difference between the proposed temporal regularization and STA in Fig. 6, should show a more definitive difference in the figure.**
>
> We zoomed into the details of Fig. 6 to compare the w/o $\mathcal L_{\rm{temp}}$​ and w/o STA variants with the leftmost column (Full Model). We provide the more-clear comparison in Fig. 8. By examining the white bounding box region in (a) and (b), we observe that there are red artifacts in the w/o $\mathcal L_{\rm{temp}}$​  variant. The artifacts are also present in the depth map , as shown in the orange region in (c) and (d). By examining the green bounding box region in (c) and (e), the geometry of  the volcano’s smoke area of the w/o STA variant is not as consistent as the full model.
>
>
> **W5: Missing citation related to panoramic image generation.**
>
> We will discuss Multi-diffusion [4], SyncDiff [5], and the approach proposed by Quattrini et al [6] in the related work.
>
> **W6: Citing the right version of the conference papers.**
>
> We will revise our draft and update the citations to the published version of the papers.
>
> **Q1: Time and GPU memory cost to generate a video.**
>
> Please find it in the general response.
>
> **Reference**
>
> [1] Unterthiner, T., Van Steenkiste, S., Kurach, K., Marinier, R., Michalski, M., & Gelly, S. (2018). Towards accurate generative models of video: A new metric & challenges. arXiv preprint arXiv:1812.01717.
>
> [2] Wang, Qian, et al. "360dvd: Controllable panorama video generation with 360-degree video diffusion model." Proceedings of the IEEE/CVF Conference on Computer Vision and Pattern Recognition. 2024.
>
> [3] Wang, J., Yue, Z., Zhou, S., Chan, K. C., & Loy, C. C. (2024). Exploiting diffusion prior for real-world image super-resolution. International Journal of Computer Vision, 1-21.
>
> [4] Bar-Tal, O., Yariv, L., Lipman, Y., & Dekel, T. (2023, July). MultiDiffusion: fusing diffusion paths for controlled image generation. In Proceedings of the 40th International Conference on Machine Learning (pp. 1737-1752).
>
> [5] Lee, Y., Kim, K., Kim, H., & Sung, M. (2023). Syncdiffusion: Coherent montage via synchronized joint diffusions. Advances in Neural Information Processing Systems, 36, 50648-50660.
>
> [6] Quattrini, F., Pippi, V., Cascianelli, S., & Cucchiara, R. (2025). Merging and Splitting Diffusion Paths for Semantically Coherent Panoramas. In European Conference on Computer Vision (pp. 234-251). Springer, Cham.

---

> ### Author Response · Authors · 2024-11-25
> **Kindly Requesting Comments**
>
> Dear Reviewer jzFB,
>
> Thank you once again for your review. As the deadline for the author-reviewer discussion is drawing near, we are looking forward to hearing your thoughts.
>
> We have addressed all your questions with additional experiments and clarifications:
>
> - We have adopted the FVD & KVD metrics to evaluate the animating phase.
> - We have extended the testing set with 32 extra scenes and will scale it up to 128 scenes.
> - We have clarified the straight-forward baseline you suggest has already been included in the main text, and we also tested the results at 4K resolution using the suggested super-resolution techniques. We have explained why the straight-forward baseline performances moderately on this task.
> - We have provided a clearer visualization of Fig. 6 and added more explanation regarding the ablation study of the lifting phase following your suggestion.
> - We have added the three papers you mentioned to the related works section.
> - We plan to check the papers cited in our work and update the citations for papers published in conference proceedings.
> - We have provided the time and gpu cost to generate a video.
>
> As the discussion period is coming to an end, we would greatly appreciate any additional feedback you might have. If our responses have clarified your understanding of our paper, we sincerely hope you can raise the rating.
>
> Thank you again for your effort in reviewing our paper.
>
>
> Best regards,
>
> Authors of Paper 1955

---

> ### Comment · Reviewer_jzFB · 2024-11-25
> **Response to rebuttal**
>
> Thank you for sharing the additional experiments and revised writing during the rebuttal period. I have checked it, as well as other reviews. However, I still have some unresolved concerns.
>
> 1. In the ablation study of the animating phase, the paper does not mention what the "motion" metric represents or how it was measured. I suggest adding an explanation and analysis of this metric. If "motion" is an important metric, why weren’t the results for the baseline, 3D-Cinemagraphy, measured? Please establish consistent evaluation criteria and measure all methods accordingly.
>
> 2. In the updated Tab. 1 and Tab. 2, shouldn't the 4K4DGen values for Q-Align (VQ) be the same? Please check this again.
>
> 3. I still cannot visually distinguish the differences in the examples shown in Fig. 6 and Fig. 8. What exactly do the authors mean by the "Artifact in RGB"? I think changing the example videos might help. I would like to ask the authors and other reviewers if they truly think the volcanic eruption case is a good example to illustrate the ablation study. Since there are no quantitative measurements in the ablation study about lifting phase, if the qualitative differences are not significant, the contribution the authors claim may not feel justified.
>
> 4. In my opinion, specifying the exact conference or journal name where the cited paper was published is the simplest task. I find it a bit difficult to understand why this hasn’t been updated yet.
>
> I understand that this project demonstrates good results. However, from the perspective of a "paper," the evaluation and ablation of the contributions claimed by the authors are neither accurate nor thorough. The ICLR 2025 Author Guide also emphasizes the importance of reproducibility in papers, but current inadequate and imprecise evaluations harm reproducibility. Additionally, small details play a significant role in determining the completeness of a paper, and in this regard, it still seems to be lacking.

---

> > ### Author Response · Authors · 2024-11-25
> > **Thank you! Working on them, and please stay tuned**
> >
> > Dear Reviewer jzFB,
> >
> > We are very thankful for your detail-oriented, constructive critiques of our work! They've been helpful.
> >
> > These are actively under work. Please allow us one day (max) before updating again & reporting back.
> >
> > Stay tuned!

---

> ### Author Response · Authors · 2024-11-26
> **Response to Reviewer jzFB**
>
> Dear Reviewer jzFB,
>
> We are thankful for your detailed comments. Please find the response below.
>
> **Q1. Definition of Motion, and Comparison with baseline methods**
> - **Definition of motion**. We refer to the video generation field [1] and adapt it for measuring the generated scene dynamics. The motion metric is calculated by averaging the magnitude of the computed optical flow cross frames. In detail, given $T+1$ frames of a video {$ I_k\in \mathbb{R}^{3\times h \times w} | k=0,1,...,T$}, its motion score presented in Table 2 is calculated as following (we have also added the explaination of motion score in the appendix):
>
>   - Calculating the optical flow magnitude between every two adjacent frames $F_k = \rm{opticalFlowMagnitude}(I_{k-1}, I_{k}) \in \mathbb{R}^{h\times w}, k=1,...T$.
>   - For each optical flow magnitude frame $F_k$, we average the magnitude over all pixels to get the per-frame motion score $\alpha_k=\sum_{i=1}^{h}\sum_{j=1}^{w}\frac{F_k^{i,j}}{h\times w}$ at the $k^{\rm{th}}$ frame.
>   - We get the final motion score $\alpha$ by averaging the per-frame motion scores over all frames $\alpha = \sum_{k=1}^T \frac{\alpha_k}{T}$. We included the definition in the appendix.
>
> - **Why does motion not serve as the evaluation metrics (in Table 1)?**
>   - We adopt the similar optical flow based motion score from SVD [1] to explain that using our animating strategy of the **animating phase** within our system can effectively alleviate the problem of too small motion when directly taking the generic narrow-FoV animator as priors.
>   - However, as shown in the attached comparison video (in revised supplementary material `1-comparsion_case1.mp4`, `2-comparsion_case2.mp4`, `3-comparsion_case3.mp4`) with 3D-Cinemagraphy, we can notice 3D-Cinemagraphy  produces a large area of ghosting artifacts. These artifacts result in rapid pixel changes in the affected areas over time. Such rapid changes artificially inflate the cross-frame optical flow, leading to abnormally high motion scores. Therefore, it is not suitable to use the motion score to evaluate different methods with different animators due to the domain gap. Thus, we only utilize the motion criteria to study the effectiveness within the module of our pipeline.
>
> **Q2: Why do the Q-Align values differ between Table 1 and Table 2? What is the difference in the two evaluation protocols used?**
>
> The Q-Align (VQ) metric in Table 1 is computed using renderings from the 4D representation, assessing the performance of the **holistic 4K4DGen pipeline**. In contrast, the Q-Align (VQ) metric in Table 2 is evaluated on the animated 2D videos to compare different animation strategies within the intermediate **animation phase**.  Since the entities evaluated by Q-Align (VQ) in Table 1 and Table 2 are different, the values should be different .To clarify this distinction, we have updated the captions and naming conventions in Table 2 to differentiate it from the evaluation of the complete 4K4DGen pipeline.
>
> **Q3: Still cannot visually distinguish the differences in the examples shown in Fig. 6 and Fig. 8.**
>
> We revised Fig. 8 in the appendix and visualize key frames to make it clear. We notice the lifting phase may generate 3D artifacts when optimizing the 3D Gaussian without the constraints.
>
> “Artifact in RGB” means there are artifacts in the rendered RGB videos.  We have made a video to make a clear comparison (`4-flashing-stripes.mp4`). Please find it in the revised supplementary material.
>
>
>
> **Q4: Specifying the conference or journal name in the references**
>
> We have revised the paper accordingly and have updated the references. Thanks for your valuable suggestion.
>
> We would like to sincerely thank you once again for your time and effort in reviewing our paper.
>
> [1] Blattmann, A., Dockhorn, T., Kulal, S., Mendelevitch, D., Kilian, M., Lorenz, D., ... & Rombach, R. (2023). Stable video diffusion: Scaling latent video diffusion models to large datasets. arXiv preprint arXiv:2311.15127.

---

> ### Comment · Reviewer_jzFB · 2024-11-27
> **Response to Authors**
>
> Thank you for addressing the requested changes in such a short time. Initially, I wanted to give a high rating because this paper is technically solid and pioneers a new field. However, the experimental section was too underdeveloped to justify a high rating.
>
> **W1. The ghosting artifacts of 3D-Cinemagraphy leads to abnormally high motion scores**
>
> The authors said that "we can notice 3D-Cinemagraphy produces a large area of ghosting artifacts. These artifacts result in rapid pixel changes in the affected areas over time. Such rapid changes artificially inflate the cross-frame optical flow, leading to abnormally high motion scores. Therefore, it is not suitable to use the motion score to evaluate different methods with different animators due to the domain gap." This indicates that the motion score is not suitable for quantitatively evaluating video generation. If ghosting artifacts can abnormally inflate the motion score, it cannot be considered an appropriate metric for this task. In that case, the authors should refrain from using this metric. It would be better to remove it altogether. I believe this paper has pioneered an entirely new research field, and I understand that this makes evaluation challenging. However, **rather than attempting quantitative evaluation at all costs, please use a reasonable metric that adequately supports the authors' claims**. If the authors genuinely want to measure quantitative results and believe that existing metrics cannot fairly evaluate the task you are addressing, propose a reasonable metric that can convince the readers. In my opinion, if the qualitative comparison is clear, results from a user study alone would be sufficient.
>
> **Additionally, there are still essential revisions needed:**
>
> 1. Reproducibility: It would be beneficial to specify and release the 16 panorama datasets used in the evaluation. If the values included in the paper reflect experiments conducted during the rebuttal period with 32 or 128 images, please ensure that these datasets are made available later.
>
> 2. Lines 419–422: Since FID and KID are no longer being used, please remove these lines.
>
> 3. Ablation Studies - Animating Phase: If the authors truly believe that the motion score is a meaningful and appropriate metric, please add at least one sentence explaining what the motion score represents in the relevant section, and include a note directing readers to the supplementary material for a detailed explanation. If there is space, it would also be helpful to explain why 4K4DGen achieves a higher motion score in the table (e.g., "the optical flow magnitude between two adjacent frames is higher" or "the motion is globally larger and more consistent per frame compared to Animate Perspective Image"). Generally, since the motion score directly inherits the capabilities of the I2V model, if Animate Perspective Image and 4K4DGen use the same pretrained I2V model, their motion scores should be similar.
>
> 4. Tab. 1 and 2: Instead of simply labeling "Ours (holistic pipeline)" in Tab. 1 and "Ours (Animating Phase)" in Tab. 2, include at least one sentence in the main paper clarifying the distinction between these setups, as described in the rebuttal: "The Q-Align (VQ) metric in Table 1 is computed using renderings from the 4D representation, assessing the performance of the holistic 4K4DGen pipeline. In contrast, the Q-Align (VQ) metric in Table 2 is evaluated on the animated 2D videos to compare different animation strategies within the intermediate animation phase." Without this explanation, simply updating the table labels makes it difficult for readers to understand the difference between the two setups.
>
> 5. Fig. 8: The updated examples now make it clearer how inconsistencies arise when the temporal loss is excluded. However, since the left side of Fig. 8 should demonstrate temporal consistency, it would be better to include at least one additional comparable reference timestep image, as seen in Fig. 6.
>
> 6. Line 608: There seems to be a typo in the citation for "360roam: Real-time indoor roaming using geometry-aware 360° radiance fields." Please update it to SIGGRAPH.
>
> 7. Supplementary Material D.2 - Motion Criteria: Please specify which algorithm was used to measure optical flow (e.g., OpenCV or RAFT[1]).
>
> 8. Supplementary Material - Generation time and memory usage: The proceeding time per generation shared in the general response would enhance the quality of the paper if added to the supplementary materials.
>
> Items 5 and 8 are optional. Since the page limit is 10 pages, it may not be possible to include everything in the main paper. However, please prioritize and incorporate the experimental results discussed during the rebuttal period to produce a more polished paper.
>
> [1] Zachary Teed and Jia Deng, RAFT: Recurrent All-Pairs Field Transforms for Optical Flow, ECCV 2020

---

> > ### Author Response · Authors · 2024-11-30
> > **Response to Reviewer jzFB**
> >
> > Dear Reviewer **jzFB**:
> >
> > We sincerely appreciate your continued engagement and valuable contributions to improving our paper. In our latest rebuttal, we have addressed your concerns by revising the **motion metrics** through a user study and producing a more polished paper, as you suggested.
> >
> > We would be grateful if you could kindly review our response and let us know if it adequately addresses your comments. We look forward to incorporating the suggested changes and hope that our work can inspire a broader audience, thanks to your valuable input.
> >
> > Best regards,
> >  Authors of #1955

---

> > > ### Comment · Reviewer_jzFB · 2024-12-01
> > >
> > > Thank you for sharing the results. Since all my concerns have been resolved, I will raise my rating from 3 to 8. Great work, and thank you for your effort.

---

> > > > ### Author Response · Authors · 2024-12-01
> > > > **Response to Reviewer jzFB**
> > > >
> > > > Dear Reviewer **jzFB**:
> > > >
> > > > We are pleased to hear that our responses have addressed your concerns. We truly appreciate your constructive comments throughout the review process, which have greatly helped in improving our work.
> > > >
> > > > Best Regards,
> > > >  Authors of #1955

---

> ### Author Response · Authors · 2024-11-28
> **Response to Reviewer jzFB**
>
> Dear Reviewer **jzFB**,
>
> We are thankful for your suggestions. Please find the response below.
>
> **W1. Evaluation of “motion” for the 4D generation task.**
>
> To better evaluate the “motion” for the 4D generation task, we have revised the evaluation metric by incorporating **user study**, focusing on the naturalness and amplitude of motion. Please refer to **Sec D.2** in the revised paper for details.
>
> - Motion's naturalness: the motion of the generated views should be natural to human’s understanding, avoiding abrupt pixel changes across frames.
> - Motion's amplitude: the motion trajectory of the **scene's subjects** should have **adequate and realistic** magnitude.
>
> As shown in the revised **Tab.1**, 4K4DGen consistently outperforms the baseline method in terms of motion naturalness and amplitude metrics, demonstrating the efficacy of the proposed method.
>
> - **Comparison with 3D-Cinemagraphy.**
>  | Method  |  Amplitude (UC) ↑  | Naturalness (UC) ↑|
>  | ------------- |:-------------:|:-------------:|
>  | 3D-Cinemagraphy (zoom-in) |        $29.4\%$          |     $19.7\%$   |
>  | 3D-Cinemagraphy (circle)     |        $32.0\%$          |     $21.1\%$   |
>  | Ours (holistic pipeline)                               |     $ 38.6\%$       |       $59.2\%$ |
>
>
> Furthermore, we also evaluated the motion’s naturalness and amplitude in the **ablation study** , as presented in **Tab. 2** in the revised paper.
>
> - **Different Animation Strategies in the Animating Phase.**
>  | Method  |  Amplitude (UC) ↑  | Naturalness (UC) ↑|
>  | ------------- |:-------------:|:-------------:|
>  | Animate Pano. |        $26.8\%$          |     $17.8\%$   |
>  |   Animate Pers. |    $32.4\%$          |     $39.3\%$   |
>  | Ours (Animating Phase)                               |      $ 40.8\%$        |      $42.9\%$ |
>
> It proves that our proposed animating strategy significantly improves the motion quality on 4K panorama while alleviating the “small motions” issue.
>
>
>
>
> ---
> ---
>
> **1.More details of Reproducibility and release the evaluation dataset.**
>
> We provide the initial paper’s 16 panoramas (`Initial-16-panoramas`) and the rebuttal period’s 32 panoramas (`extra-32-panoramas`) in the revised supplementary material. Furthermore, we will make our panorama datasets and related code publicly available.
>
> **2. Delete the statements of FID and KID.**
>
> Thanks for your suggestions. We have removed the statements of FID and KID metrics from the paper.
>
> **3.  Add additional explanations to clarify why the proposed method yields the best motion results.**
>
> - Following your suggestion, we replaced the motion metric with a user study to better explain the small motion issue observed in the "animating entire panorama" baseline. The user study focuses on evaluating two aspects: **motion naturalness** and **motion amplitude**. Participants are asked to select one or more videos from the provided set that exhibit noticeably greater amplitude or better naturalness. The ratio of votes received by each method to the total votes serves as a metric to assess motion amplitude and naturalness.
>
> - In the motion comparison of “Animate Pano. Image”, “Animate Persp. Image” and “4K4DGen”, we observe that high-resolution pano. animating would result in small motion due to the domain gap, while per-perspective animating cannot produce motion “across perspectives” like us.  By contrast, 4K4DGen capitalizes the generative ability from perspective animating priors while enabling cross-view consistent motion between different perspectives, which achieves the best motion naturalness and amplitudes among all the settings.
> We have added it  to the revised paper (see Table 2’s caption).
>
> **4. Revise the captions and explanations  for Table 1 and Table 2.**
>
> We have added the suggested explanation to the revised paper.
>
> **5.In Fig. 8, it would be better to include additional reference timestep images.**
>
> We have incorporated the initial frame as the reference timestep image, and we have revised Fig. 8 based on your suggestions. We appreciate your valuable feedback.
>
> **6. A typo in the citation for "360 Roam”.**
>
> We have fixed it in the revised paper. Thank you for your suggestion.
>
>
> **7. More details about the original Motion Criteria in the Appendix.**
>
> The optical flow is computed via the calcOpticalFlowFarneback function in OpenCV, which is an offline, and dense optical flow prediction algorithm. Since we have revised the motion evaluation, we have removed the related descriptions.
>
> **8. Add Generation time and memory usage to the supplementary materials.**
>
> We have appended the "Experimental Details - Sec D.1" section to the supplementary materials.
>
>
> We would like to sincerely thank you once again for your time and effort in reviewing our paper.

---

### Official Review · Reviewer_2Ek6 · 2024-11-03

**Soundness:** 3
**Presentation:** 4
**Contribution:** 3
**Rating:** 8
**Confidence:** 4

**Summary:**

The paper proposes an end-to-end solution for generating realistic 4D panoramic videos from static panoramas at 4K resolution for the first time. The paper uses pretrained image to video models to convert the static 2D panoramas to videos and then dynamically (with spatio-temporal alignment) lift them to 3D space resulting in the 4D content.
The paper demonstrates the appropriateness of using a spherical latent space for the task and how the panoramic denoiser utilizes that. The lifting uses a geometry alignment model (pre-trained depth estimator) and optimize DreamScene360's objective with an additional temporal loss.

The framework achieves high fidelity generations, both, visually and quantitatively. The user-study also aligns well with the results.

**Strengths:**

1. Novelty: This is the first paper that allows seamless (spatio-temporally aligned) 4K resolution generation of 4D content.
2. The spherical latent space operation alleviates a lot of visual artifacts and this paper is evidence that it is suitable for Panoramic generation tasks.
3. The model components, intuition and math is well grounded.
4. The fidelity of results is promising. Quantitative metrics suggest the same. (The resulting panorama videos provided on the supplementary page are extremely visually pleasing!)

**Weaknesses:**

1. Why are Efficient4D and 4DGen not used as baselines that potentially fail on Panoramas as there is no spatial-temporal alignment mechanism there? It would make the paper much stronger.
2. It would have been nice to see comparisons for the 4D lifting phase to see how the proposed is better than existing methods like: OmniNeRF. This way the speed of the proposed method can also be highlighted as a strength of the paper.
3. More visual results in the main text would be nice.

**Questions:**

My main concern is the lack of baselines in the paper but the problem statement, conceptualization and results are very convincing hence the accept rating.

Question(s):
Q1. What is the processing times per generation on the A100 used? I would also be interested in knowing how memory intensive it is.
Q2. If more baselines are added (see weaknesses), I would be willing to upgrade my recommendation score to 8.

Suggestion(s):
* L295: Fix typo: biuld -> build
* The space is low but can the authors somehow manage to split the Evaluation paragraph starting on L407? (split at the each underlined term)

---

> ### Author Response · Authors · 2024-11-22
> **Response to Reviewer 2Ek6**
>
> **W1: Why Efficient4D and 4DGen are included as baseline?**
>
> The Efficient4D [1] and 4DGen [2] frameworks differ significantly from our approach, making direct comparisons challenging. The distinctions lie in twothree main areas: (1) Task: Efficient4D and 4DGen focus on generating dynamic objects from text or images, which is fundamentally different from our objective of generating large-scale panoramic 4D scenes. (2) Camera Settings: In Efficient4D and 4DGen, the cameras used for optimization and training are inward-facing, capturing the entire object from various angles. In our task, the cameras are outward-facing, each capturing different parts of the overall scene.
>
> Nevertheless, we attempted to adapt the 4D object generation framework 4DGen [2] to our settings. We provided the model with a perspective of the volcano from our ‘volcano’ scene. The results, shown in Fig. 9, indicate that the generated object differs significantly in form from our scene outputs, making meaningful comparison difficult.
>
> **W2: Adding OmniNeRF as lifting phase baseline?**
>
> Thank you for your constructive suggestion. Following your advice, we adopted OmniNeRF [3] as the baseline for the lifting phase. As it is a single frame lifting method, we feed OmniNeRF [3] the RGB and depth frame that come from our method as input and optimize the NeRF. We observe that it has two main drawbacks: (1) It struggles to optimize 4K panoramas, for it is very slow to optimize. It took over 8 hours to optimize a $1024 \times 512$ panorama, while our method only took 19 minutes. (2) Its optimized geometry is not as sharp as ours, which is evidenced by Fig. 10. We add it as our baseline and make a comprehensive comparison in the final version of OmniNeRF [3].
>
> **W3: Adding more visual results in the main text.**
>
> Thank you for your suggestion. We will reorganize the paper and include more qualitative results in the main text.
>
> **Q1: Time and memory cost to make a single generation?**
>
> Please find it in the general response.  We will incorporate this into the Experiments section.
>
> **Q2: Adding more baselines?**
>
> For the lifting phase, the comparison with OmniNeRF [3] is presented in the answer of W2, and we will incorporate these results in our revision. Regarding the SDS-based 4D object generation pipelines [1, 2], as noted above, directly comparing them with our framework is difficult. Nevertheless, we will include a more comprehensive discussion of these methods in the related work section.
>
> **Q3: Typos and Splitting the ‘Evaluation’ paragraph.**
>
> We have corrected the typos in the revision and will reorganize the draft by splitting the paragraph you mentioned. Thank you for your suggestion.
>
> **Reference**
>
> [1] Pan, Z., Yang, Z., Zhu, X., & Zhang, L. (2024). Fast dynamic 3d object generation from a single-view video. arXiv preprint arXiv:2401.08742.
>
> [2] Yin, Y., Xu, D., Wang, Z., Zhao, Y., & Wei, Y. (2023). 4dgen: Grounded 4d content generation with spatial-temporal consistency. arXiv preprint arXiv:2312.17225.
>
> [3] Hsu, C. Y., Sun, C., & Chen, H. T. (2021). Moving in a 360 world: Synthesizing panoramic parallaxes from a single panorama. arXiv preprint arXiv:2106.10859.

---

> > ### Comment · Reviewer_iVVu · 2024-11-26
> >
> > Thank you for answering! It solves all my questions when first reading the paper.

---

> ### Author Response · Authors · 2024-12-02
> **Response to Reviewer 2Ek6**
>
> Dear Reviewer **2Ek6**,
>
> We are thankful for your constructive comments. Please find our response below. If you find our responses helpful, we sincerely hope the reviewer could kindly consider raising the score accordingly.
>
> ### **W1: Adding more baseline methods to compare with 4K4DGen?**
> To further address your primary concern, we adopt two types of baseline methods: **4D Object Generation** and **4D Scene Generation**, to compare with the proposed method. Note that the panoramic 4D generation is still underexplored due to the scarcity of annotated data and the lack of well-trained prior models tailored for panorama format. As a result, we find that existing methods cannot achieve similar quality as 4K4DGen in this task.
>
>
> - **4D Object Generation methods**
>
> Following your suggestion, we have devoted substantial engineering efforts to adapt the 4D object generation framework **4DGen** [1] to our camera settings, as shown in the qualitative results in **Fig. 9** of the revised paper.
> It demonstrates that recent 4D object generation methods struggle to generate scene-level content due to inherent domain gaps between objects and scenes. Our method overwhelmingly outperforms 4DGen [1] in terms of quantitative evaluations (e.g., Image Quality, Image Aesthetics, and Video Quality) and qualitative evaluations.
>
> - **4D Generation  Methods for Scene**
>
> We also construct two 4D scene generation baselines:
>
> (1) We equip **LucidDreamer** [2] with our animator. We follow the authors' setting, using ZoeDepth and its inpainting model (Stable Diffusion) to expand invisible views for each timestamp, and then we use our backbone animator to animate and optimize the Gaussians.
>
> (2) We employed a very recent 4D scene generation technique **DimensionX** [3].  We use the authors' model configuration, employing its lora ('orbit_left') model to generate novel views and 3D structures. Since the DimensionX's T-Director is currently unavailable, we leveraged the same backbone animator from our approach to provide temporal guidance for its 4D representation [4] in the 4D generation stage.
>
> Compared with existing methods, including **4DGen**, **LucidDreamer**, and **DimensionX**, our method consistently achieves higher quantitative results, demonstrating the efficacy of the proposed method.
>
>  | Method  |  Q-Align (IQ) ↑  | Q-Align (IA) ↑| Q-Align (VQ) ↑ |
>  | ------------- |:-------------:|:-------------:|:-------------:|
>  | 4DGen [1] (object) |     $0.19$     |          $0.20$   |     $0.29$   |
>  | 3D-Cinemagraphy (zoom-in) |    $0.47$      |  $0.38$    |   $0.57 $  |
>  | 3D-Cinemagraphy (circle)     |        $0.48$         |  $0.40 $     |  $0.58 $ |
>  | LucidDreamer + Our Animator   |   $0.44$ | $ 0.41 $      |      $0.58$ |
>  | DimensionX +  Our Animator   |  $ 0.55$       | $0.42$       |    $0.60 $ |
>  | $\textbf{Ours (holistic pipeline)} $         |   $ \textbf{0.66}$       |  $\textbf{0.44}$    |  $\textbf{0.62}$ |
>
>
> We will include these experiments (both quantitative and qualitative results) in the revised manuscript. We will also consider including more recent baselines in the final version.
>
> We would be grateful if you could kindly review our response and let us know if it adequately addresses your concerns. We would like to sincerely thank you once again for your time and effort in reviewing our paper.
>
> Best regards, Authors of #1955
>
> $\newline$
>
> ---
> [1] Yuyang Yin, Dejia Xu, Zhangyang Wang, Yao Zhao, and Yunchao Wei. 4DGen: Grounded 4d content generation with spatial-temporal consistency. arXiv preprint arXiv:2312.17225, 2023.
>
> [2] Jaeyoung Chung, Suyoung Lee, Hyeongjin Nam, Jaerin Lee, and Kyoung Mu Lee. Luciddreamer: Domain-free generation of 3d gaussian splatting scenes. arXiv preprint arXiv:2311.13384, 2023.
>
> [3] Sun, W., Chen, S., Liu, F., Chen, Z., Duan, Y., Zhang, J., & Wang, Y. (2024). DimensionX: Create Any 3D and 4D Scenes from a Single Image with Controllable Video Diffusion. arXiv preprint arXiv:2411.04928.
>
> [4] Yang, Z., Gao, X., Zhou, W., Jiao, S., Zhang, Y., & Jin, X. (2024). Deformable 3d gaussians for high-fidelity monocular dynamic scene reconstruction. In Proceedings of the IEEE/CVF Conference on Computer Vision and Pattern Recognition.

---

### Official Review · Reviewer_iVVu · 2024-11-03

**Soundness:** 3
**Presentation:** 3
**Contribution:** 3
**Rating:** 8
**Confidence:** 3

**Summary:**

This paper proposes a novel framework to generate 4K4G panoramic video rendered from 3D Gaussian representation. The key technical insights is to use pre-trained 2D perspective camera to denoise the panorama by first rendering it into perspective image patch and then fusing it back into the sphere space. Once a panoramic video is created, a lifting process built on single image depth prediction method is to turn each frame into 3D Gaussian representation so that we can navigate inside the panorama. Both quantitative and qualitative experiments verified the effectiveness of each technical components and overall the framework achieves impressive video generation results.

**Strengths:**

1. As far as I know, this is the first paper to generate high-resolution 4D panorama videos, which may have many applications in VR.

2. The proposed method is technically solid and easy to follow. The consistent panoramic animation part is similar to prior works that use 2D diffusion model to generate panorama but the paper replaces 2D diffusion models with 2D video diffusion model. The dynamic panoramic lifting part is also reasonable.

3. While evaluating generative models is always a difficult problem and there is no prior works solving this problem, this paper still manages to build a set of benchmarks and a reasonable baseline to show the effectiveness of the proposed framework, which should be appreciated.

5. The paper is well-written and easy to follow.

**Weaknesses:**

I am not an expert in using diffusion models for video generation and I haven't followed recent progress in that direction. The consistent panoramic animation part looks solid and reasonable to me. I only have some minor questions that are not necessarily the weakness of the paper.

1. Eq. (3): While this equation is similar to how we generated static panorama using diffusion models for 2D perspective image, I wonder whether that will cause different Gaussian noise distribution at the overlap regions. Since we are essentially average the Gaussian noise from two denoised latent code, will that cause the variance of the noise at the overlap region to be smaller compared to non-overlapping area? Will that cause artifacts in image generation?

2. My second question is how significant the dynamic panoramic lifting can help us render 3D video. Given that all the examples shown in the paper have very large-scale scene and the perturbation $\alpha$ is very small in training, I would guess whether we lift the panorama into 3D or not may not make a big difference and approximating the whole 3D structure with a sphere may give us similar results. I assume the comparison is shown in Figure 6(c) but it is very difficult for me to see the difference.

3. Also I wonder why don't we choose recent dynamic Gaussian works to model the video. I assume that is a more systematic way to handle this problem and may give us more temporal consistent view synthesis results.

**Questions:**

My major question is on the necessities of the dynamic panoramic lifting and whether that actually improves the quality when we move the camera and look around. In the submitted video, we can see a little bit camera movements but I wonder if that can be mimiced but simply textured the panorama to a large enough sphere. Also when handling the dynamic scene, why not consider all the dynamic Gaussian papers and instead use static Gaussian with a temporal constraints? If we use dynamic Gaussian representation, will that cause any differences?

---

> ### Author Response · Authors · 2024-11-22
> **Response to Reviewer iVVu**
>
> **W1: In Eq. (3), would the variance decrease in the overlapped region?**
>
> Yes, the variance does decrease in the overlapped region. However, we do not observe significant artifacts in the generated video, which is also consistent with findings from widely used tile-based approaches in image generation tasks [1, 2, 3].
>
> That said, it is interesting to consider how the decreased variance impacts the sampling procedure, both in image and video diffusion models. We plan to further investigate this effect in future work.
>
> **W2: Necessities of the dynamic panoramic lifting phase?**
>
> Lifting the panoramic video into a true 3D structure allows dynamic novel view rendering with stereoscopic effect for VR/AR/XR applications, which provides a more immersive user experience than putting the video on a large sphere. We provide the renderings of a lifted 3D scene where a user walked along a street in Fig. 11. Notice the roof highlighted by green bounding boxes in (a) and (b). When the user walks nearer, the more area of the roof will be observed. It is hard to implement such effect without the lifted 3D structure.
>
> **W3: Why don't we use dynamic 3D Gaussians to model the video?**
>
> We opted not to use dynamic 3D Gaussians in this work because our goal is to achieve high visual quality for an immersive user experience. We empirically found that dynamic 3d gaussians [4] leads to more blurry results than our final approach. For example, it struggles to model the special objects like the smoke or the fire of the volcano. This finding is also  validated in the L4GM paper [5]. However, we acknowledge that dynamic 3D Gaussians are significantly more computationally efficient. We plan to further investigate dynamic 3D Gaussians to explore potential trade-offs between visual quality and computational efficiency.
>
>
> **Reference**
>
> [1] Bar-Tal, O., Yariv, L., Lipman, Y., & Dekel, T. (2023, July). MultiDiffusion: fusing diffusion paths for controlled image generation. In Proceedings of the 40th International Conference on Machine Learning (pp. 1737-1752).
>
> [2] Shi, Y., Wang, P., Ye, J., Long, M., Li, K., & Yang, X. (2023). Mvdream: Multi-view diffusion for 3d generation. arXiv preprint arXiv:2308.16512.
>
> [3] Feng, M., Liu, J., Cui, M., & Xie, X. (2023). Diffusion360: Seamless 360 degree panoramic image generation based on diffusion models. arXiv preprint arXiv:2311.13141.
>
> [4] Wu, G., Yi, T., Fang, J., Xie, L., Zhang, X., Wei, W., ... & Wang, X. (2024). 4d gaussian splatting for real-time dynamic scene rendering. In Proceedings of the IEEE/CVF Conference on Computer Vision and Pattern Recognition (pp. 20310-20320).
>
> [5] Ren, J., Xie, K., Mirzaei, A., Liang, H., Zeng, X., Kreis, K., ... & Ling, H. (2024). L4GM: Large 4D Gaussian Reconstruction Model. arXiv preprint arXiv:2406.10324.

---

### Official Review · Reviewer_pfwC · 2024-11-04

**Soundness:** 3
**Presentation:** 4
**Contribution:** 3
**Rating:** 6
**Confidence:** 4

**Summary:**

This paper presents a method by which a static pano image is animated to produce a video pano, then subsequently lifted to 3D using 3DGS.  The authors claim the following contributions:
* 4K4DGen, a method for lifting a static pano to dynamic 3DGS
* A panoramic denoiser which generates consistent animated perspective views
* A dynamic panoramic lifting which transforms the video pano to dynamic 3DGS

**Strengths:**

* compelling use case
* qualitative results look good
* quantitative metrics show improvements over baselines
* well written

**Weaknesses:**

* The animations each have small spatial extent.  I didn't see an example that illustrated that you could have many of these overlapping perspective animations and for a large scale animation to emerge.  e.g., It would be much more impressive if the authors could show the car in the desert where the car drives around the center of project.

* Some of the generated videos have significant artifacts.  For example, the fireworks in the third row of the supplemental has fireworks that don't move and speckled artifacts that are static.

* The motion isn't always globally consistent.  For example, I'm not sure the clouds in row 7 of the supplemental make sense.

* I think overall, the idea of applying this existing model to perspective patches is neat, but I'd be concerned this work would be quickly superseded by one where the diffusion model was fine-tuned on 360 videos.

* The lifting phase doesn't seem sufficiently evaluated.  It would have been nice to have something to inspect in the supplemental.

**Questions:**

* Have the authors considered using a more recent depth model than MiDaS?  It's pretty old at this point.

---

> ### Author Response · Authors · 2024-11-22
> **Response to Reviewer pfwC**
>
> **W1: Animations have a small spatial extent?**
>
> Our method already has the ability to generate animations across the panorama (see the clouds in the second row of the supplementary materials). Generating large, arbitrary object motion relies on the motion prior of the backbone animator when applied to perspective views. We will further investigate the capability of producing larger object motion, leveraging advanced and controllable animators in future works.
>
> **W2: In the firework scene, some regions do not move and there seems to be specked artifacts?**
>
> The non-moving region lies outside the motion mask provided by the user, thus it is not animated.
>
> **W3: Clouds in row 7 of the supplemental do not make sense?**
>
> There are distortions inherent in equirectangular panoramas, making panoramic videos appear unnatural when viewed directly in their equirectangular projection. We provide videos rendered from several perspective views, which should appear more coherent. Additionally,  existing generative animators cannot always ensure physical correctness, which occasionally leads to results that are not physically realistic.
>
> **W4: Would it be quickly superseded by one where the diffusion model was fine-tuned on 360 videos?**
>
> In the short term, training panoramic video models to produce high-quality 4K videos suitable for immersive VR/AR/XR experiences is difficult, primarily due to the lack of high-quality panoramic video datasets. For instance, the recent WEB360 [1] dataset presents several challenges for training a robust generative model: (1) Small scale: it contains only around 2,000 panoramic videos, (2) Low resolution: the videos are at $1024\times512$ resolution, which is significantly lower than 4K, and (3) Low quality: the dataset includes many low-quality, web-collected videos, such as those with watermarks or mosaic. With this limited data, it becomes extremely challenging to train a model capable of generating high-quality panoramic videos like ours.
>
> **W5: The lifting phase seems not sufficiently evaluated？**
>
> The lack of ground truth geometry makes it difficult to evaluate the lifting phase alone. Following previous scene-level generation pipelines in 3D settings [2, 3, 4], we use renderings from various perspective views as a reference to evaluate the geometry quality, i.e., better renderings imply better underlying geometry. Additionally, we report conventional image quality metrics for the lifting phase. Typically, the PSNR metric for randomly selected views ranges between 38–40 dB, indicating a high level of scene reconstruction accuracy.
>
> **Q1: Using more recent depth estimators instead of MiDas?**
>
> We will employ the depth-anything model [5] to generate the initial depth for our framework. We will include quantitative results with this model in our experiment.
>
> **Reference**
>
> [1] Wang, Q., Li, W., Mou, C., Cheng, X., & Zhang, J. (2024). 360dvd: Controllable panorama video generation with 360-degree video diffusion model. In Proceedings of the IEEE/CVF Conference on Computer Vision and Pattern Recognition (pp. 6913-6923).
>
> [2] Zhou, S., Fan, Z., Xu, D., Chang, H., Chari, P., Bharadwaj, T., ... & Kadambi, A. (2025). Dreamscene360: Unconstrained text-to-3d scene generation with panoramic gaussian splatting. In European Conference on Computer Vision (pp. 324-342). Springer, Cham.
>
> [3] Yu, H. X., Duan, H., Herrmann, C., Freeman, W. T., & Wu, J. (2024). WonderWorld: Interactive 3D Scene Generation from a Single Image. arXiv preprint arXiv:2406.09394.
>
> [4] Chung, J., Lee, S., Nam, H., Lee, J., & Lee, K. M. (2023). Luciddreamer: Domain-free generation of 3d gaussian splatting scenes. arXiv preprint arXiv:2311.13384.
>
> [5] Yang, L., Kang, B., Huang, Z., Xu, X., Feng, J., & Zhao, H. (2024). Depth anything: Unleashing the power of large-scale unlabeled data. In Proceedings of the IEEE/CVF Conference on Computer Vision and Pattern Recognition (pp. 10371-10381).

---

### Author Response · Authors · 2024-11-22
**General Response**

We would like to firmly express our gratitude to all reviewers for their insightful and constructive
Comments, and appreciate the comments like “compelling use case and qualitative results look good” (pfwC), “impressive generation results and have many applications in VR ”(iVVu), “high fidelity generations, both, visually and quantitatively” (2Ek6), and “great spatial and temporal consistency” (jzFB)

Thank you very much for your time and valuable suggestions! We would be eager to hear from you if there are any further concerns we have not yet addressed.


**The proceeding time of per generation**

* Animating phase:
We provide the time and GPU cost to animate a single video at different resolutions in the following table.


 |Resolution | GPU Usage (GB) | Time Cost (Minutes) |
| ------------- |:-------------:|:-------------:|
 | $1024\times512$ |         9.38             |               0.89                |
 | $2048\times 1024$|          29.31            |            2.99                    |
 | $4096\times 2048$ |         73.41             |             22.31           |

* Lifting phase:
We provide the time and GPU cost to lift a single frame at 4K, 2K, and 1K resolution. The results are shown in the following table.

|Resolution | GPU Usage (GB) | Time Cost (Minutes/frame) |
| ------------- |:-------------:|:-------------:|
 | $1024\times 512$ |           7.59           |                    19       |
 | $2048\times 1024$ |         12.07             |                22        |
 | $4096\times 2048$ |          31.27            |                33      |

* Conclusion:
The computational cost increases rapidly with the resolution, making 4K generation highly challenging.  We will put more engineering effort to accelerate the generation pipeline in the future.

---

### Meta-Review · Area_Chair_FADv · 2024-12-14

**Metareview:**

The paper presents an approach to lift 2D panorama images to 3D and animate them.
It builds on existing work for animation and depth estimation, and proposes an adoption to panoramic data.
To handle scarcity of training data, the paper leverages 2D diffusion models.

The reviewers appreciated the novelty of the proposed work and the qualitative results.
The paper was considered well written. The method is quantitatively evaluated against reasonable baselines.

The main concerns were raised by jzFB who remarked a limited set of experimental results and those being an insufficient basis for measuring FID and KID. Further concerns regarded ablation experiments and comparisons to baselines. All of these concerns were addressed during the discussion phase with the reviewers finally recommending to unanimously accept the paper.

**Additional Comments On Reviewer Discussion:**

- more recent dynamic Gaussian work for modeling video [iVVu], addressed through discussion
- missing evaluation of lifting phase, comparison to OmniNeRF [pfwC, 2Ek6,jzFB], addressed through additional experiments & discussion
- Lifting necessary? [iVVu] adressed through discussion
- Baselines Efficient4d, 4DGen [2Ek6], addressed through discussion and additional results
- baseline animate panorama [jzFB], addressed
- meaningless FID & KID evaluation [jzFB], addressed
- limited number of results [jzFB]
- related work [jzFB], addressed
- lack of details, imprecise evaluation [jzFB], addressed

---

### Decision · Program_Chairs · 2025-01-22

Accept (Spotlight)